# Continental-scale analysis of shallow and deep groundwater contributions to streams

Danielle K. Hare [1,2✉], Ashley M. Helton[1,3], Zachary C. Johnson [4], John W. Lane[5] & Martin A. Briggs [5]

Groundwater discharge generates streamflow and influences stream thermal regimes. However, the water quality and thermal buffering capacity of groundwater depends on the aquifer source-depth. Here, we pair multi-year air and stream temperature signals to categorize 1729 sites across the continental United States as having major dam influence, shallow or deep groundwater signatures, or lack of pronounced groundwater (atmospheric) signatures. Approximately 40% of non-dam stream sites have substantial groundwater contributions as indicated by characteristic paired air and stream temperature signal metrics. Streams with shallow groundwater signatures account for half of all groundwater signature sites and show reduced baseflow and a higher proportion of warming trends compared to sites with deep groundwater signatures. These findings align with theory that shallow groundwater is more vulnerable to temperature increase and depletion. Streams with atmospheric signatures tend to drain watersheds with low slope and greater human disturbance, indicating reduced stream-groundwater connectivity in populated valley settings.

[1] Department of Natural Resources and the Environment, University of Connecticut, Storrs, CT, USA. [2] Volunteer, U.S. Geological Survey, Earth Systems Processes Division, Hydrogeophysics Branch, Storrs, CT, USA. [3] Center for Environmental Sciences & Engineering, University of Connecticut, Storrs, CT, USA. [4] U.S. Geological Survey, Washington Water Science Center, Tacoma, WA, USA. [5] U.S. Geological Survey, Earth System Processes Division, Hydrogeophysics Branch, Storrs, CT, USA. ✉email: danielle.hare@uconn.edu

Groundwater discharge zones establish active stream–groundwater hydrologic connectivity through the advective exchange of water. As a critical contributor to streamflow generation, groundwater discharge influences water quantity and quality throughout stream networks, especially during seasonal low flows and dry conditions[1]. Many streams host ecologically important 'groundwater-dependent ecosystems'[2], yet these habitats face growing threats from climate change and groundwater contamination[1,3,4]. Aquatic organisms are particularly susceptible to shifts in thermal regimes because they have life cycles that rely on annual thermal cues[5] and metabolic rates influenced by stream temperature[6].

The relatively stable thermal regimes of some groundwater discharge zones can buffer stream temperatures against long-term air temperature trends and short-term hot and cold extremes[2]; therefore, groundwater discharges can provide important stream channel thermal refuges and refugia for sensitive aquatic organisms such as salmonid fishes[7,8]. However, in response to climate change and land development, streams and rivers have recently shown widespread warming[9,10]. Observed stream warming trends are spatially heterogeneous due in part to spatially variable groundwater contributions to streamflow[11]. Thus, effective watershed management will require a process-based characterization of groundwater contribution to streamflow[12] at ecologically relevant scales to predict future stream thermal regimes.

The magnitude, spatial distribution, and source-flow path characteristics of groundwater discharge can control the physical characteristics of individual streams[8,13,14] and whole stream networks[15]. Characterizing the depth of contributing groundwater is particularly important for understanding broad-scale responses of stream ecosystems to land development and climate change[16] for three main reasons: first, groundwater depth is associated with annual thermal stability as natural surface temperature fluctuations are prominent within the shallow aquifer but quickly attenuate with depth[13]. Deeper groundwater (defined here as greater than approximately 6 m from the land surface) shows little annual thermal variability relative to shallow groundwater[17] that flows through the near-surface portion of the 'critical zone'[18]. Therefore, groundwater discharge can either impart stability (deep groundwater) or variability (shallow groundwater) on atmospheric-driven stream thermal regimes. Hydrogeologic climate simulations support this definition, as water tables below 5 m have shown decoupling from surface energy balances[19]. Second, shallow groundwater is inherently more sensitive to land-use changes[20] and surface contamination[21–23]. Thus, effective watershed management may have a different urgency depending on the depth of contributing groundwater. Also, naturally, deep and shallow groundwater tend to have different chemical profiles[24–26], which has important implications for surface water quality and stream ecosystem function including delivery of legacy contaminants[15]. Third, shallow groundwater can be directly depleted via transpiration[27], irrigation withdrawals[28], and is more vulnerable to seasonal water table drawdown during dry periods while discharge from deeper groundwater sources is more seasonably stable[29]. This depth-dependent effect can affect stream water transit times and catchment water balance, emphasizing the importance of parsing shallow versus deep contributing groundwater flow paths[24].

Though understanding the implications of climate change and land development for stream ecosystems requires quantifying the magnitude and source-depth of groundwater discharge, we lack efficient and broadly applicable methods to characterize source groundwater depth. Most hydrologic techniques for evaluating the physical properties of groundwater discharge are labor-intensive and not spatially and temporally scalable[30]. More efficient methods, such as stream water temperature sensitivity linear regression analyses[31] or physically based hydrograph separation techniques[32] do not directly differentiate groundwater source-depth. Inference of groundwater source-depth is possible using water chemistry end-member mixing[33] or water isotopic data[34], but these analyses cannot inherently specify shallow groundwater flow paths without additional hydrologic characterization, and are time and resource-intensive.

In the absence of groundwater discharge, annual stream water temperature signals are often well coupled to seasonal variation of local air temperature[35]. A departure from this coupling in terms of seasonal magnitude and timing is characteristic of influence from varied depth groundwater discharge[8] or dam operation[36]. Discharge of shallow groundwater to streams has physical properties closely tied to seasonally dynamic air temperature and precipitation, quickly responding to short-term perturbations such as hot, dry summers[37]. Discharge from deep groundwater sources does not tend to respond to anomalous weather years but is sensitive to long-term climate trends at extended time scales ranging from decadal to centennial[16,38,39].

In this work, we used a newly refined methodology to classify 1729 stream sites across the continental United States as having shallow or deep groundwater signatures, lacking a pronounced groundwater signature, or having major dam influence, based on publicly available multi-year air and stream water temperature records and metadata. Our analysis harnesses the relatively high annual variability in shallow groundwater temperatures and the stability of deep groundwater temperatures to identify characteristic paired air and stream water annual temperature signal relations. We used our classification to (1) compare our annual temperature signal-based categorization to baseflow indices, (2) explore continental spatial patterns and landscape drivers of groundwater discharge characteristics, and (3) evaluate how stream temperature is changing over time (14–30 years) among streams with varied source-depth of groundwater discharge. We present an unprecedented broad-scale inference of groundwater discharge contribution to streams that will inform more accurate predictions of stream responses to changing climate and land use conditions.

## Results and Discussion

**Continental classification.** We used paired air and stream water annual temperature signal relations to broadly classify stream and river sites with atmospheric (i.e., lacking a pronounced groundwater signature), deep groundwater, shallow groundwater, or major dam signatures across the continental U.S. Our sites represent a broad range of stream sizes encompassing 1st to 9th order (median: 3rd order) across 21 of the 25 U.S. physiographic provinces (categorized based on large-scale geomorphology; Supplemental Table 1). We used multi-year annual temperature signals as a diagnostic tool because they are less susceptible to variable flow and weather than other stream temperature-based groundwater discharge metrics that rely on short-term thermal variance[40]. Streams below major dams have complex, management-influenced annual thermal regimes[36] and are not explored in detail here.

For streams with substantial groundwater discharge, the amplitude and phase of paired annual air and stream water temperature signals decouple in distinctive ways. At sites with a deep groundwater signature, the annual stream temperature signal is highly damped compared to air—quantified by the stream water/air amplitude ratio—but the signals are approximately in-phase. Groundwater discharge from shallow flow paths causes variable stream temperature signal damping, but uniquely shifts the timing of the annual stream water temperature signal later relative to the annual air temperature signal—quantified by the time-forward phase lag. This characteristic phase lag propagates into stream water from adjacent shallow aquifers,

whereas deeper groundwater flow paths have a highly attenuated annual temperature signal and thus do not influence the stream water signal phase[8]. For our broad-scale analysis, we assigned categories of shallow and deep groundwater signatures according to paired air and stream water annual signal metrics of amplitude ratios and phase lags based on previous analyses[8,40,41]. We assigned sites that either had phase lags of greater than 40 days, which is not an expected outcome of even extreme shallow groundwater discharge mixing with stream water[8], or are within 25 km downstream of major dams, as sites with major dam signatures. Of the 1729 sites we categorized, 305 sites met this dam criterion and are removed from the groundwater signature analysis.

Sites classified as having pronounced groundwater signatures are common in this national dataset. We found that of the 1424 sites analyzed for groundwater signatures, groundwater substantially influences the annual thermal regimes of 39% ($n =$ 556). We classified 47% ($n = 264$) of these sites as having deep groundwater signatures, and 53% ($n = 292$) as having shallow groundwater signatures (Fig. 1). The average amplitude ratio is 0.54 ($\sigma = 0.10$) for sites with deep groundwater signatures and 0.59 ($\sigma = 0.18$) for sites with shallow groundwater signatures. The air to stream water annual signal phase lag averaged 16.6 days ($\sigma = 6.6$ days) for sites with shallow groundwater signatures and 3.8 days ($\sigma = 3.4$) for sites with deep groundwater signatures. In contrast, the average amplitude ratio for sites with atmospheric signatures is better coupled to annual air temperature at 0.85 ($\sigma = 0.12$) with a negligible average phase lag of 2.3 days ($\sigma = 2.7$ days) that is not significantly different than zero phase lag.

Deep and shallow groundwater contributions to streamflow are not mutually exclusive, often a spectrum of flow path depths contributes to streamflow[42], but our analysis derives which signature is dominant. The distribution of annual signal metrics within our groundwater contribution categories indicate that our thresholds that define the groundwater signature categories occur near natural breaks (Supplementary Fig. 1), indicating alignment with potential groundwater-driven separations of underlying populations in the data.

We compared our temperature-based approach for classifying groundwater contribution to streamflow data by using multi-year baseflow regression analysis for the subset of sites that had concurrent streamflow records ($n = 554$) (Fig. 2). Specifically, we calculated the baseflow index (BFI), an estimate of the ratio of baseflow to total streamflow based on the annual stream hydrograph, as it is one of the few current methods for quantifying relative groundwater contributions to streamflow efficiently at broad scales[32]. As may be expected, sites with atmospheric thermal signatures had significantly lower BFIs (median—0.69) than sites with either shallow groundwater (median BFI – 0.79) or deep groundwater (median BFI—0.86) signatures (Fig. 2). This result aligns with theory that the primary driver of baseflow throughout river networks is groundwater discharge.

BFI varies among groundwater contribution categories; streams with shallow groundwater signatures have significantly lower

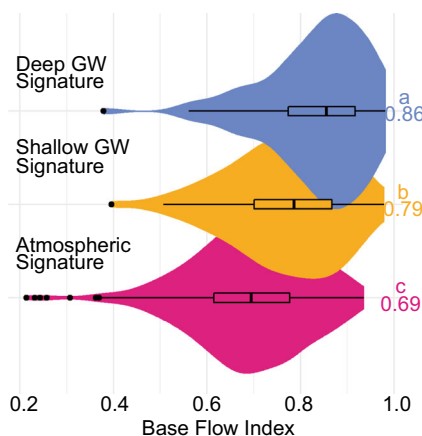

**Fig. 2 Categorical groundwater (GW) signatures compared to baseflow index (BFI).** Letters indicate significance at $p < 0.05$ reported alongside median BFI. Counts of each category are atmospheric signature (pink) $n =$ 401; shallow groundwater GW signature (yellow) $n = 71$; deep groundwater GW signature (blue) $n = 82$. Boxplots center line is the median and box limits are the upper and lower quartiles.

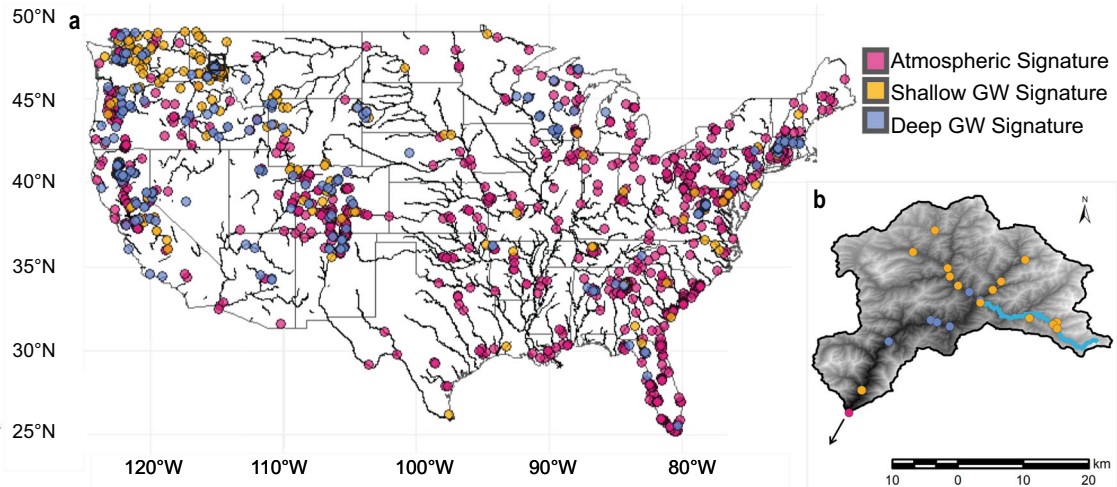

**Fig. 1 Spatial distribution of stream sites by categorical groundwater signature.** Categorical groundwater (GW) signatures derived from annual paired air–stream water temperature signals **a** across the continental United States and **b** within a single watershed, the North Fork of the Clearwater River-Lake Creek watershed, Idaho—Montana, USA (Hydrologic Unit Code HUC10 – 1706030701). Lake Creek stream is highlighted. Across the United States, counts of each category are atmospheric signature (pink) $n = 868$; shallow groundwater GW signature (yellow) $n = 292$; deep groundwater GW signature (blue) $n = 264$. Legend descriptions are maintained between **a** and **b**. Base map **a** was generated from R package 'maps' version 3.3.0 and the Nation Hydrography Dataset[70] **b** was created from 7.5-minute ground surface elevation data courtesy of the U.S. Geological Survey.

BFIs than those with deep groundwater signatures. This observation supports site-specific research that found shallow groundwater sources are less reliable for generating baseflow at seasonal timescales[29,37]. Shallow (less than 6 m depth) aquifer flow paths drain a relatively small groundwater reservoir that is highly sensitive to seasonally dynamic recharge rates and transpiration[27], and are therefore less-reliable generators of stream baseflow. In contrast, deep groundwater flow from larger reservoirs is generally sustained throughout the year[42,43] at a more constant rate[44], increasing the average baseflow index in streams dominated by deeper groundwater discharge. This result highlights that effective water resource and aquatic habitat management in a changing world should consider both groundwater connectivity and the source-depth of groundwater discharge.

**Spatial patterns and physical drivers.** Our results demonstrate that the spatial distribution of groundwater contributions to streamflow is complex across the continental United States, but large-scale spatial patterns emerge (Fig. 1a). Physiographic provinces with the highest percentage of deep groundwater signatures are often associated with those expected to have productive aquifers, such as glaciated terrains (e.g., 31% of sites in New England have a deep groundwater signature) or sedimentary bedrock (e.g., 27% of sites in the Colorado Plateau have a deep groundwater signature) (Supplementary Table 1). Physiographic provinces that have a high proportion of streams draining steep mountainous terrain with thin soil coverage generally have a higher percentage of shallow groundwater signatures (e.g., Northern Rocky Mountains—74% of sites have shallow groundwater signatures) (Supplementary Table 1). Thus, landforms and geologic structures are likely, in part, controlling the spatial patterning of groundwater contribution to streams across the United States. Yet, within regions, there is substantial heterogeneity in groundwater signatures. For example, in the Cascades-Sierra Mountains, 38% of sites have shallow groundwater signatures, and 32% of sites have deep groundwater signatures. This observation is likely in part because of the geologic variation between the High Cascades (younger, highly fractured volcanic bedrock) and Western Cascades (shallow soils, and abundance of clay)[37]. Also, within the Coastal Plain province (eastern coastline of the United States from Massachusetts to Mexico), while 91% of sites have an atmospheric signature, sites with shallow and deep groundwater signatures do occur in isolated areas such as the Floridian Section that is dominated by karst aquifers (Fig. 1, Supplementary Table 1). Indeed, atmospheric, shallow, and deep groundwater signatures co-occur within all eight physiographic regions and within 18 out of 21 physiographic provinces considered in our study. Previous research has shown broad-scale mapping of expected stream water– groundwater connectivity characteristics which can be inferred with a combination of physiography and climate, a concept supported with relatively sparse BFI analysis[43]. Because low-cost stream temperature measurements are currently being performed at thousands of publicly available sites nationally, paired air and stream water temperature signal-based analysis offers a highly scalable approach to provide additional specificity regarding groundwater discharge dynamics, refining broad-scale zonation of stream water–groundwater connectivity.

Among physiographic regions, local watershed characteristics likely also play an important role influencing groundwater discharge to streams[45]. Overall, sites with shallow groundwater signatures tend to have higher watershed slopes than sites with atmospheric or deep groundwater signatures (Fig. 3a). We hypothesize that watersheds with higher slopes are more likely

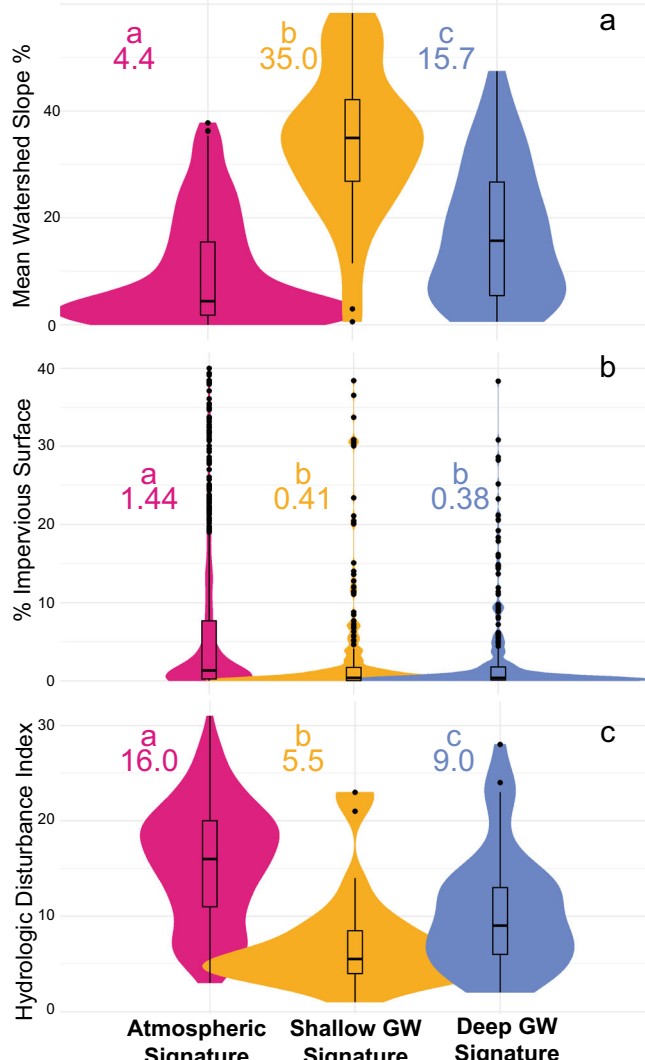

**Fig. 3 Watershed properties for groundwater (GW) signature categories.** **a** Mean slope of the watershed draining to each site. **b** Percent impervious surface from the year 2011 of the local catchment draining to each site. Y-axis is truncated at 40% impervious surface, which removed 44 outliers from atmospheric signature and 5 outliers from shallow groundwater GW signature categories. **c** The Hydrologic Disturbance Index for each site based on the GAGES-II dataset[52,63]. Higher values indicate more disturbance. For **a** and **c** site counts of each category are atmospheric signature (pink) $n = 277$; shallow groundwater GW signature (yellow) $n = 40$; deep groundwater GW signature (blue) $n = 51$. Boxplots center line is the median and box limits are the upper and lower quartiles. For **b** site counts of each category are atmospheric signature (pink) $n = 831$; shallow groundwater GW signature (yellow) $n = 275$; deep GW groundwater signature (blue) $n = 246$.

to have a shallow depth to bedrock, which is a known driver of near-surface hillslope groundwater flow to streams[46]. Yet, our results show that strong connectivity of streams and shallow groundwater occurs in environments beyond smaller, steep headwater streams, such as areas with shallow confining layers[47]. Sites with shallow groundwater signatures drain larger watersheds (median 153 km²; Q1–Q3: [17 km², 2131 km²]), have higher streamflow (median 13 m³ s⁻¹), and have a greater range of streamflow (Q1–Q3: [2 m³ s⁻¹, 98 m³ s⁻¹]) than sites with deep groundwater signatures (watershed size: 65 km²; Q1–Q3: [18 km², 616 km²]; streamflow 2 m³ s⁻¹; [0.4 m³ s⁻¹, 10 m³ s⁻¹])

suggesting shallow groundwater signatures occur across a wide spectrum of hydrogeologic settings that may not be predicted by current conceptual models of baseflow generation.

Heterogeneity in groundwater signatures exists even at the sub-watershed scale. For example, at the North Fork Clear Water—Lake Creek watershed in Idaho, USA (Fig. 1b), sites within the steep headwaters are dominated by shallow groundwater signatures while sites along the mainstem river valley are largely characterized by deep groundwater signatures, with the outlet of the watershed shifting to an atmospheric signature. This watershed represents an important habitat for a range of cold-water salmonid species[48]. Interestingly, a major tributary (Lake Creek, highlighted in Fig. 1b) was moved to the list of impaired waters in 2010 by the Idaho Department of Environmental Quality for elevated temperature criteria violations[48]. Without explicit consideration of groundwater dynamics, this impairment was attributed to a slight reduction in canopy shading (4%) compared to the local shade optimal target. However, of the four sites we investigated in upper Lake Creek watershed, one main stem stream and three tributaries, all are classified as having shallow groundwater signatures of greater than 15-day phase lags. These large phase lags suggest dominance of the annual thermal regime by shallow groundwater, and we speculate that the previously observed warm stream impairment is due in part due to warming of shallow groundwater. Consideration of local to regional groundwater responses to climatic and watershed modifications is crucial yet often overlooked in stream temperature predictions, which can mislead future projections and produce less effective mitigation strategies when ignored. The multi-scale heterogeneity of groundwater contribution to streamflow within and among physiographic regions and individual watersheds provides the impetus for higher spatial resolution regional characterization for targeted cold-water species management.

**Human drivers of stream/groundwater disconnection**. Human alterations can also influence the spatial patterns of groundwater connectivity and discharge to streams[49]. Our results demonstrate that streams with atmospheric signatures tend to occur in local catchments (area directly draining to a river segment, excluding any upstream contribution[50]) with a higher percentage of impervious surface area (Fig. 3b). Sites with atmospheric signatures also tend to have a higher "Hydrologic Disturbance Index" (HDI), which is a more holistic metric of human influence derived from seven anthropogenic watershed modifications, not including percent impervious cover[51,52] (Fig. 3c). The median HDI score for atmospheric signature sites is 16 and a maximum of 31. Sites with pronounced deep groundwater signatures have a median watershed HDI of 9 and shallow groundwater signature sites have a median HDI score of 5.5 (Fig. 3c). This discrepancy in HDI scores between groundwater categories may result in part from the fact that human disturbance is more immediately influential to shallow groundwater dynamics, and therefore fewer streams in such disturbed basins show shallow groundwater discharge signatures, compared to more resilient deeper groundwater. One of these seven HDI parameters is groundwater withdrawal, which has been shown to have immediate effects on streamflow generation, especially within areas reliant on irrigation, and is generally projected to increase in the future to offset droughts[53]. We hypothesize that in addition to pumping, the relative lack of sites with groundwater signatures observed in this study in more disturbed landscapes is a result of the many human landscape modifications that reduce groundwater discharge to streams and rivers. These impacts occur either directly through groundwater withdrawal or indirectly through impervious cover

and stormwater infrastructure that saps shallow groundwater and diverts precipitation to streams, reducing infiltration and aquifer recharge. Therefore, streams within watersheds with high human modification, predominantly in lowlands, are likely to have lower groundwater connectivity and be more susceptible to warming, though recent research suggests that extreme low flows may be buffered along urban corridors due to infrastructure-based recharge[54]. Understanding how human modifications alter groundwater discharge dynamics across the U.S. will therefore involve disentangling how urban development interacts with geology and landscape features.

**Stream temperature temporal trends**. Quantifying the thermal stability of streams influenced by groundwater discharge is essential in predicting the effects of climate change on stream networks. The capacity of stream water temperature to be buffered against a warming world depends in part on the source depth of groundwater discharge[55], and high groundwater connectivity is often invoked as a primary driver of persistent cold-water habitat[8]. Indeed, of the 184 sites that had long-term contiguous temperature records (ranging 14 to 30 years), we found that sites with deep groundwater signatures had a substantially smaller proportion of significant positive temperature trends than sites with shallow groundwater or atmospheric signatures (Fig. 4). More than half of the long-term sites with atmospheric signatures ($n = 132$) have stream water temperatures that are increasing over the last 14 to 30 years ($n = 70$), ranging from 0.01 to 0.09 °C yr$^{-1}$ (μ: 0.04 °C yr$^{-1}$). Similarly, for long-term sites with shallow groundwater signatures ($n = 29$), we found that 45% have stream water temperatures that are increasing with rates of warming ranging from 0.01 to 0.1 °C yr$^{-1}$ (μ: 0.04 °C yr$^{-1}$). The rates of warming for sites with shallow groundwater signatures and atmospheric signatures are consistent with previously reported stream water warming trends[9,10].

In contrast to sites with shallow groundwater signatures, 52% of sites with deep groundwater signatures had stable stream water temperature regimes (Fig. 4a,b). This finding underscores the strong thermal buffering capacity of deep groundwater discharge and the likely greater resistance to climate warming of groundwater-dependent and cold-water habitat sourced by deep compared to shallow groundwater. The six deep groundwater signature sites with significant warming trends had rates ranging from 0.01 to 0.05 °C yr$^{-1}$ (μ: 0.01 °C yr$^{-1}$). Sites with deep groundwater signatures also showed the greatest proportion (22% of sites) of significant cooling trends. Although stream cooling trends appear counterintuitive under climate change, they have also been identified in previous work[56], and may be due to localized changes in winter precipitation patterns[57].

The difference in thermal buffering capacity of streams dominated by shallow versus deep groundwater discharge has been predicted by modeling efforts for individual watersheds[29,37,55]. Our empirical results confirm these predictions and expand evidence to sites across the United States. We recognize that there are confounding factors that influence long-term stream temperature, notably discharge variability. Therefore, streams fed by shallow groundwater could warm at a faster rate in part because of drought conditions or groundwater withdrawal (e.g., for irrigation) lowering groundwater levels, which disproportionately affects shallow groundwater[28].

The disparity between long-term stream temperature trends of sites with shallow versus deep groundwater signatures also occurs during the summer season, when cold water fishes are most often thermally stressed. Over 70% of sites with shallow groundwater signatures show significant summer season warming trends compared to 43% of sites with deep groundwater and 61% of sites with atmospheric signatures (Fig. 4c). These seasonal

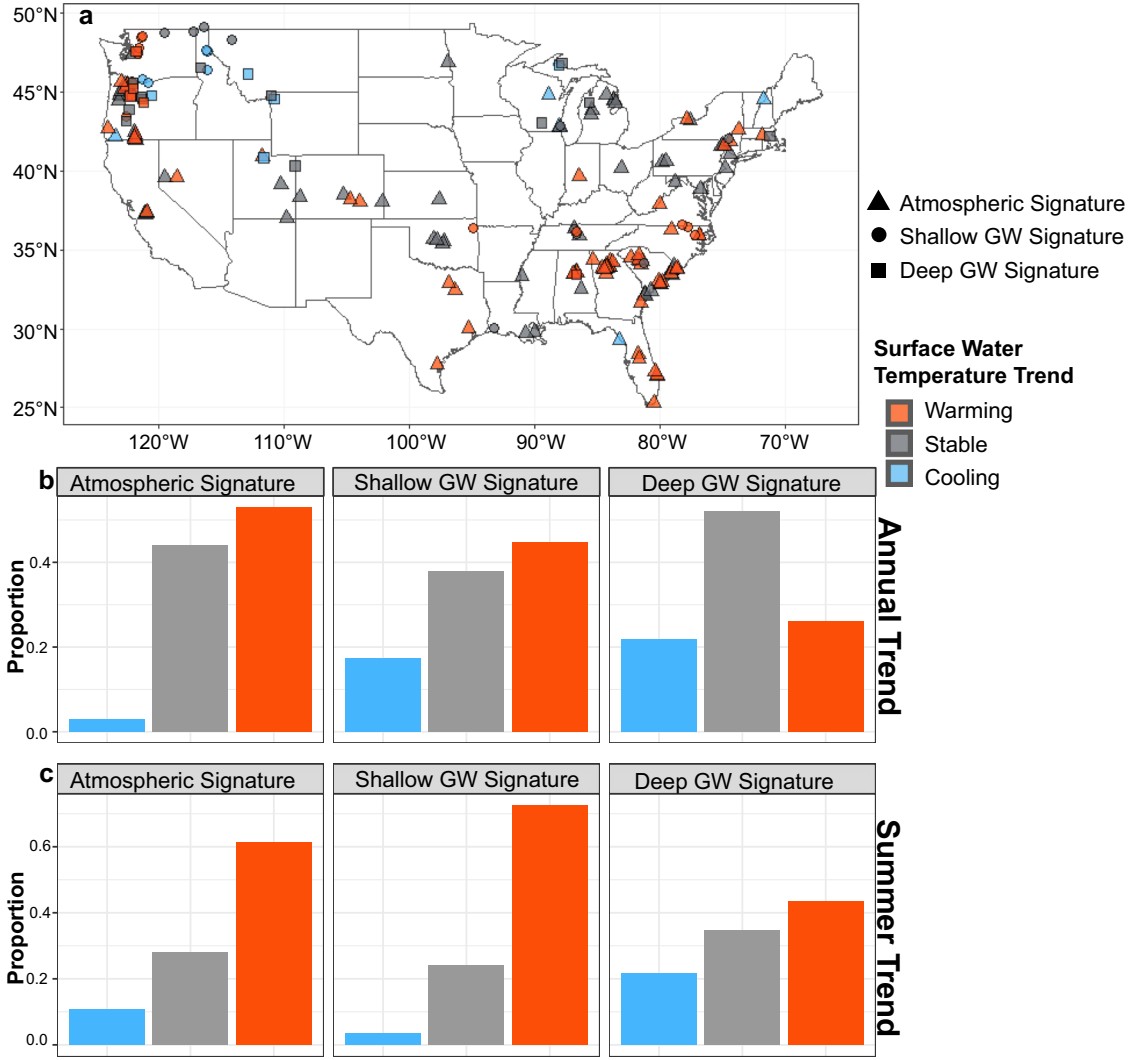

**Fig. 4 Stream water temperature trends based on average monthly values for 14–30 years of data post 1990. a** Spatial distribution of stream water temperature annual trends across the United States by groundwater (GW) signature category. Base map was generated from R package 'maps' version 3.3.0. **b** The proportion of sites with long-term annual temperature increasing (warming (red), $p < 0.05$), decreasing (cooling (blue), $p < 0.05$) monotonic trends, or stable condition (gray) ($p > 0.05$) by GW signature category. **c** Similarly, the long-term temperature trends based on summer temperatures (June – August) by GW signature category. Site counts of each category in **a–c** are atmospheric signature (triangle) $n = 132$; shallow GW signature (circle) $n = 29$; and deep GW signature (square) $n = 23$.

warming trends follow the fundamental nature of the classification method, which relies on the pronounced annual temperature signals of shallow groundwater to be transferred to stream water via groundwater discharge zones. Sites with shallow groundwater signatures will be immediately sensitive to hotter summers, exacerbating thermal stress on sensitive aquatic organisms[41]. Thus, vulnerable biota within streams dominated by shallow groundwater may not only have to adapt to a warming baseline condition, but also be particularly vulnerable to the impacts of single season heatwaves. Deep groundwater is more resistant to land surface temperature changes, but still sensitive to longer-term thermal shifts at timescales tied to source flow path depth[38]. This re-emphasizes the importance of distinguishing shallow versus deep groundwater source-depth, rather than assuming streams with strong baseflow components imply thermal stability.

Groundwater discharge to streams and rivers occurs via a spectrum of source groundwater flow paths, which exerts high-level controls on streamflow, channel thermal stability, and stream water quality characteristics that are tightly linked to the source aquifer. The relative flow path depth of contributing

groundwater is particularly important for stream ecosystems; yet, until recently we lacked efficient process-based methodology to parse the relative dominance of shallow or deeper groundwater discharge to streams at broad spatial scales. Our continental-scale characterization demonstrates a framework for harnessing burgeoning publicly available air and stream temperature datasets to categorize the relative flow path depth of groundwater contribution to streams and rivers, which can inform how both hydrologic models and stream ecosystem management approaches incorporate groundwater dynamics.

**Implications of groundwater discharge source-depth.** Groundwater-dependent ecosystems have become an important consideration for watershed management decisions[1], and streams with substantial groundwater contributions are generally considered most resilient to change. Our work underscores the need for expanding the direct incorporation of groundwater discharge dynamics, especially source-flow path depth, into decision-making processes and predictive frameworks. Streams with

shallow or deep groundwater signatures were ubiquitous nationally (nearly 40% of sites) and distributed across stream sizes, U.S. physiographic provinces, and within regional sub-watersheds. Yet, regional generalizations remain uncertain at scales relevant for managing stream habitat. Although the more thermally stable streams with deep groundwater signatures tended to occur more frequently in regions with productive aquifers and in watersheds with lower slopes, they also occurred across nearly all physiographic provinces, and a range of watershed slopes and drainage areas. Human land development may explain some of the heterogeneity in groundwater connection, as we found that sites with groundwater signatures were less likely to be associated within catchments with high impervious cover or other types of human disturbance, including groundwater pumping and channelization.

Our characterization of groundwater contribution to stream-flow has important implications for understanding and predicting how streamflow and water quality respond to climate change, groundwater extraction, and watershed development. By definition, shallow aquifer flow paths with pronounced annual temperature signals are tightly coupled to seasonal temperature (and precipitation) dynamics, and our analysis shows that streams influenced by shallow groundwater are more likely to be warming over time than sites with deep groundwater signatures. Shallow groundwater discharge will then have reduced stream cooling potential in summer, particularly during anomalous seasons, when thermal refuges in marginal cold-water habitat are most needed. Our analysis also shows that streams influenced by shallow groundwater tend to have a reduced fraction of total streamflow composed of baseflow compared to deep groundwater. Thus, streams with substantial shallow groundwater contribution are more vulnerable to extreme low flows or drying from climate change-related increases in drought or evapotranspiration, or from increased groundwater extraction. The high responsiveness of shallow groundwater to land surface disturbances also suggests streams with substantial shallow stream water contributions are likely more susceptible to diffuse nutrient and other pollution additions, while deeper groundwater can perpetuate legacy watershed land uses[3] and emerging contaminants such as per- and polyfluoroalkyl substances from outside the river corridor[4]. Still, shallow groundwater dominated streams may be more responsive to short-term management actions that reduce groundwater extraction and limit land application of fertilizers and other chemicals. Thus, our analysis provides foundational knowledge to the importance of source groundwater discharge flow path depth on stream temperature, flow, and water quality. We consider this additional dimension of groundwater discharge essential to informing current stream process models and necessary to building robust predictions in a time of change.

## Methods

We classified streams by their groundwater signature based on the observed decoupling of annual air and stream water temperature signals, both in terms of amplitude and timing (phase), which is driven by the magnitude and relative source-depth (shallow versus deep) of groundwater discharge to streams[8]. Shallow groundwater is defined here as groundwater within the near-surface critical zone where annual aquifer temperature is highly variable (within approximately 6 m from land surface), and this variability is transferred to streams through groundwater discharge zones causing annual temperature signal mixing with characteristic outcomes. Thermally stable, deeper groundwater discharge serves to attenuate annual stream temperature signals but does not cause notable phase shifts, as deeper groundwater temperature signals are highly attenuated. We used this newly expanded signal processing-based methodology (explained below, see refs. [8,40]) to infer the source-flow path depth of groundwater discharge to streams based on these first principles.

We acquired publicly available data from ~4000 discrete stream water temperature stations, of which 1811 met our required data criteria of being located within 25 km of a National Oceanic and Atmospheric Administration (NOAA) air

temperature station, and having at least 2 consecutive years of temperature data collected in 2010 or after without gaps of 30 continuous days or more. This data gap criteria is supported by parallel paired air and water temperature signal analysis research[58]. Stream temperature datasets were used from three repositories: the USGS National Water Information System database (NWIS)[59], the NorWeST Stream Temperature dataset[60], and the Spatial Hydro-Ecological Decision System (SHEDS); all repositories are assumed to have internal quality assurance and quality control (QA/QC) protocol. 1729 sites met our data quality review, which are discussed in the Temperature Signal Processing Approach section below.

We acquired the paired daily air temperature record for each stream station from the Global Historical Climatology Network-Daily (GHCN-daily) Database[61] using the R package 'rnoaa'[62]. We extracted data from the two nearest NOAA stations. The nearest air station data were used first; however, if the data did not meet our criteria (75% of annual data available and 75% data overlap with paired stream temperature), then a second NOAA station, if available, was evaluated and used if the criteria were met ($n = 191$).

We linked coordinates of each stream site to the nearest National Hydrography Dataset Plus flowline common identifier (COMID) (within 250 m) and paired with the U.S. Environmental Protection Agency (EPA) Stream-Catchment (StreamCat) dataset[50] to obtain watershed land cover. We also paired NWIS sites[59] with the USGS Geospatial Attributes of Gages for Evaluating Streamflow, version II (GAGES II) dataset[63] by station identifier (ID) value to obtain distance from nearest major dam, watershed slope, and the Hydrologic Disturbance Index. The Hydrologic Disturbance Index is derived from anthropogenic disturbances within the site's watershed including the presence of major dams, change in reservoir storage from 1950 to 2009, percentage of canals, road density, distance to nearest major pollutant discharge site, estimate of fresh-water withdrawal, and calculated fragmentation of undeveloped land[51].

To categorize sites into shallow groundwater, deep groundwater, atmospheric, or major dam signatures, we designed an automated signal processing software tool in Python that fits a static sine curve to the stream water and local air temperature data and derives the paired air and stream water signal metrics of amplitude ratio and phase lag. Although some datasets were collected at sub-daily frequency, average daily values were used for both air temperature and stream water temperature input data. Based on principles described in previous studies, we excluded average daily temperature readings below 1°C from the analysis, because the paired air–stream temperature relationships decouple due to the freeze–thaw dynamics of water[35]. Also, stream values greater than 60°C were removed during analysis.

For each discrete temperature record, we fit the annual temperature cycle using a linearized static sinusoidal function (equation 1) by minimizing the root mean square error (RMSE) of the average daily temperature residuals (°C) with the Python scipy optimize curve fit module[64]. This function was chosen to most simply extract the 'average' fundamental (annual) signal from the time series and is consistent with the analysis conducted by previous studies[8,40]. The average daily root-mean-square error for both air and stream water signals at each site are provided in the Fig. 1 Source files.

$$\alpha \sin(t) + \beta \cos(t) + C \qquad (1)$$

Using the calculated regression coefficients $\alpha$ and $\beta$, we calculated the amplitude ($A$; equation 2) and the phase ($\phi$ in radians; equation 3) of each signal. January 1 was defined as 1/365.

$$A = \sqrt{\alpha^2 + \beta^2} \qquad (2)$$

$$\phi = \arctan\left(\frac{\beta}{\alpha}\right) \qquad (3)$$

We defined the groundwater signature categories by the paired air and stream water signal metrics, which are amplitude ratio ($A_r$) and phase lag ($\Delta\phi$). We calculated $A_r$ by dividing the annual stream water signal amplitude by the annual air temperature signal amplitude; $\Delta\phi$ is calculated as the difference between the phase of the annual stream water temperature signal and that of the air temperature signal and converted from radians to days (d) using 365 divided by $2\pi$. A positive phase lag indicates the number of days the fitted stream water signal is delayed with respect to the fitted air temperature signal.

Negative phase lags imply that stream water temperature responds to atmospheric thermal input faster than air, which is not logical for natural stream systems (except those influenced by geothermal heating). As a result, within the dataset we explored negative phase lags ($n = 454$, mean of −4). Negative phase lags greater than 10 days ($n = 25$) were dropped from the analysis as these data were associated with heavily managed stream flows as indicated by visual inspection of the stream temperature patterns or highly variable winter air temperature data that were not well captured by the fitted sine curve. Negative phase lags between 0 and −10 days ($n = 429$) are still included within the dataset but set to 0 for calculations. These data and multi-day atmospheric signature phase lags were attributed to inherent imprecision of signal fitting to natural data, as other studies that use this same method did not show any negative phase lags when using streamside air signals[40,41]. Because the classification analysis only utilized parameters $\alpha$ and $\beta$, and not $C$, we assumed altitude differences between air temperature and stream water temperature sampling location did not have substantial influence on the amplitude ratio or phase lag.

We categorized sites as having an atmospheric, shallow groundwater, or deep groundwater signature by identifying 'conservative' threshold values of $A_r$ (0.65) and $\Delta\phi$ (10 days) that parsed only sites with pronounced groundwater signatures (Supplementary Fig. 1). These threshold values were chosen based on previously presented stream and groundwater annual signal-mixing theory, process-based modeling, and field data[8,40]. Specifically, we developed $A_r$ and $\Delta\phi$ thresholds using evidence from three well-studied systems, the Quashnet River, Cape Cod, Massachusetts[8], Shenandoah National Park, Virgina[41], and the Olympic Experimental State Forest, Washington[40]. The hydrogeology of the Quashnet River has been extensively characterized[65,66], indicating streamflow is dominated by deep groundwater discharge that at times makes up close to 100% of total streamflow. Using a dynamic sinusoidal regression technique, Briggs et al.[8] found that $A_r$ ranged from approximately 0.49 to 0.63 over a 3-year period with varied climatic conditions. Thus, we chose a threshold of 0.65 to indicate a deep groundwater signature for our study. It is likely that $A_r$ values up to approximately 0.75 also indicate substantial deep groundwater influence, but with less certainty. Other physical factors such as channel confinement, aspect, and shading could affect $A_r$, but to date no published work that we are aware of indicates these factors could explain $A_r < 0.65$ without the influence of groundwater. However, we hypothesize that these factors are likely to change the distance downstream these annual signals can be detected. All $A_r$ values less than 0.4 were manually checked for a major dam within 30 km upstream of the site by visual inspection. Extensive field data collected at Shenandoah National Park, a region known to be dominated by shallow bedrock conditions, indicates an average $\Delta\phi$ of 11 days, and conceptual mixing models of stream and groundwater annual temperature signals from Shenandoah headwater streams indicate a $\Delta\phi$ of about 10 days or greater when shallow groundwater discharge contributes at least 25% of total streamflow[8]. Therefore, for our analysis we used the threshold phase lag of 10 days to identify sites with a shallow groundwater signature. $A_r$ and $\Delta\phi$ thresholds may vary among watersheds and regions and thus can and should be modified based on additional information about individual watersheds for more precise, localized analyses. However, for the purposes of our analysis these thresholds represent conservative values applied across broad spatial scales. Sites with atmospheric signatures in our dataset had an $A_r$ between 0.65 and 1.1. Sites with deep groundwater signatures had an $A_r$ of 0.05 to 0.65. Sites with amplitude ratio values greater than 1.1 were removed as these extremes likely reflected poor pairings between the air and stream water station data, or measurement error. Because there are different numbers of sites within each groundwater signature category, we used a modified comparison of means for unbalanced designs for all statistical comparisons[67].

For sites within the USGS NWIS dataset[59], stream discharge data for 554 stream water sites were available for the same time record as the analyzed temperature dataset. We calculated baseflow index (BFI) for the 554 stream discharge stations to provide a direct comparison between typically used hydrograph separation methods and our temperature-based methods. We used the 'bfi' function within USGS-R 'DVstats' package version 0.3.4 to calculate percent baseflow for each site by averaging the percent daily baseflow (daily baseflow discharge divided by total daily flow) over the time period of the temperature record.

We analyzed a subset of our stream water temperature records for monotonic 14-year to 30-year trends (January 1990—December 2019). This record length was chosen to account for the El Niño-Southern Oscillation (ENSO) period, which is three to seven years, thus the minimum length of record (14 years) would encapsulate at least two full cycles. We recognize that these time series are short when accounting for Pacific Decadal Oscillations; however, our results indicate there is not a distinction between sites located in the western United States and the rest of the sites. Of the 1424 stream sites without major dam signatures, 197 sites had stream water temperature records with greater than 14 years of complete year records (i.e., greater than 75% of daily average temperature data) within a 30-year time span (1990–2019). Of the remaining sites, we removed a total of 13 sites manually due to data inconsistencies, such as anomalous value sets and managed patterns determined by visual inspection; therefore, 184 sites were analyzed for long-term stream temperature trends. We determined non-parametric Theil–Sen regression slopes for both annual and summer (June–August) time periods using the TheilSen function from the R package 'openair'[68], which allows for the seasonality of average monthly data to be detrended and is robust against outliers. Previous studies have stated the Theil Sen approach is comparable to a simple linear regression method when analyzing long-term stream temperatures[9]. We used the monthly averages to reduce autocorrelation and the 'deseason' option of the function to account for potentially important seasonal temperature influences such as changes to snowmelt.

## Data availability

The datasets generated during and/or analyzed during the current study are available in the USGS National Water Information System (NWIS) repository (http://waterdata.usgs.gov/nwis); the NorWest Stream Temperature repository (https://www.fs.fed.us/rm/boise/AWAE/projects/NorWeST.html); the Spatial Hydro-Ecological Decision System (SHEDS) repository (http://db.ecosheds.org/); and the NOAA Daily Global Historical Climatology Network (GHCN-Daily) repository (https://www1.ncdc.noaa.gov/pub/data/ghcn/daily/). Watershed parameters are from two publicly available datasets: the USGS data release for GAGES-II (https://water.usgs.gov/GIS/metadata/usgswrd/XML/gagesII_Sept2011.xml) and EPA StreamCat dataset (https://www.epa.gov/national-aquatic-resource-surveys/streamcat). Source data are provided with this paper.

## Code availability

Mathematical algorithms used for the analysis are presented within the text and provide sufficient information for data replication. Signal processing automation code is available on GitHub[69].

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

## Acknowledgements

We acknowledge funding from NSF-DEB grant #1655790 to AMH and NSF-EAR grant #1824820 to AMH and MAB, along with support from the USGS Toxic Substances Hydrology Program. We thank J.A. Barclay and L.E. Koenig for sharing important insight and guidance with method development. We also thank B.L. Kurylyk and H.I. Essaid for their input and advice on the draft manuscript. The work by DKH was partially done while serving as a Volunteer for Science with the U.S. Geological Survey. Any use of trade, firm, or product names is for descriptive purposes only and does not imply endorsement by the U.S. Government.

## Author contributions

D.K.H., M.A.B., A.M.H., and J.W.L. conceived the study and conceptualized the analysis. D.K.H. acquired datasets and conducted the research. M.A.B. and Z.C.J. provided methodological expertise and guidance. D.K.H. drafted the manuscript and all authors contributed to manuscript text and revisions.

## Competing interests

The authors declare no competing interests.
