## [Peer Review File · Nature Communications]

REVIEWER COMMENTS

Reviewer #1 (Remarks to the Author):

Nature Communications manuscript NCOMMS-20-22175 describes a novel study to assess groundwater (GW) contributions to streams across the contiguous USA. The study is impressive in both its geographic scope and its key findings, namely a) that shallow groundwater driven streams may be more sensitive to climate change than previously suspected and b) that traditional baseflow separation approaches may not be capable of revealing finer-scale differences in groundwater contribution needed for effective management of temperature sensitive streams. I also appreciated the authors' thorough and interesting interpretation of geographic trends in GW contributions in relation to established hypotheses regarding factors driving deep vs. shallow groundwater inputs, and the section on stream temperature temporal trends is particularly important. I therefore see this manuscript as a potentially significant and original contribution to improved understanding and management of streams threatened by climate change, and feel that the results will be of broad appeal to river managers and researchers working at the intersection of climate change, river ecology and hydrology/hydrogeology. This being said, despite these largely positive comments, I do have some moderate suggestions/improvements (listed below) that I feel would further strengthen a revised version of the manuscript; these relate primarily to clarifying questions regarding their methodology and potential limitations of their approach. Provided they are able to account for these changes, I would support the manuscript's publication in Nature Communications.

General comments

- Many of the paired T_w/T_a sites that the authors have used in their analyses comprise less than 3-4 years' (recent) data; I think it is important to demonstrate that the use of these relatively short (and recent) time series have not biased the results. One simple way to do this would be to compare the stability of the amplitude/lag metrics from a handful of shorter time series to nearby stations with longer records to ensure that the shorter series are not biased. The use of short time series also has implications for baseflow separation analyses, as 5 years' data is often suggested as a minimum time series for a robust BFI calculation. It would therefore be good to incorporate some discussion regarding this and potential limitations.
- Another slight concern is the lack of validation regarding the attribution of deep or shallow GW contributions (for sites outside of the Quashnet and the Shenandoah). I appreciate that gathering isotope/geochemical/borehole data for a multitude of rivers is beyond the scope of this manuscript, and I therefore stress that I do not feel that this limitation is a major issue. However, it would nonetheless be useful to present some information regarding how well the findings (in terms of deep vs. shallow GW contributions) conform to previously established groundwater contribution data (from isotopes separation or similar) for other select rivers, if such data exist. If these data are not available, I think that at minimum it would be important to acknowledge this as a potential limitation.
- I liked the discussion regarding the heterogeneity of GW contributions within individual watersheds, but I do wonder to what extent differences in phase lag and amplitude between upstream and downstream locations within a same watershed could actually be explained by variability in energy fluxes at the air-water interface and the advection of heat from upstream, rather than due to local changes in GW contribution. For example, could the transition from open terrain to riparian tree cover (and the change in insolation that this entails) be responsible for a similar type of change in the T_a/T_w amplitude signal to that of a change in GW contribution? It would be good to include some discussion regarding this point.
- The decisions regarding the use of the A_r and $\Delta\phi$ values for classification as shallow vs deep

groundwater are nicely justified in the text, but I would be interested to know what the results look like if these thresholds were relaxed somewhat. Have the authors done any sensitivity testing in this regard? Also, is there a case to be made for illustrating rivers along a 'gradient' of deep to shallow groundwater contribution (similar to Johnson et al. 2020, STOTEN), rather than using hard classification thresholds?

- The 'Implications' section is good but it seems overly long and a little repetitive. I therefore feel that it could be cut down by about 1/3. Please go through this section and remove any detail that has already been covered in the preceding text.

Specific comments

L38: Missing word: '...zones ARE an important driver...'

L50: I'm not sure if this clause ('Earth's most recent geological time period...') is necessary.

L55: Suggest including a hyphen in 'source-flow path characteristics'

L63: Move reference 14 to end of this sentence ('...low permeability bedrock').

L102: Please include a one-sentence explanation of what a 'physiographic province' is for non-American readers?

L109-112: It would be helpful to spell out why shallow groundwater contributions shift the phase lag of the Ta/Tw signal.

L129-130: This sentence ('Within a stream...') seems to be missing something. Please rewrite for clarity.

L132: Put 'thus not normally distributed' in parentheses.

Figure 1: Please consider colouring the actual rivers/river reaches rather than the individual points. I know that technically speaking, the computed GW contribution class only applies to the exact point location where the Tw logger is installed, but I think that it would make the figure much more visually appealing if the rivers themselves were coloured.

L164: Remove 'annual temperature signal based' from this sentence – it makes it difficult to follow.

L167: Missing word: 'We hypothesize THAT this difference...'

L172: Please rephrase this sentence. The word 'lumping' is a bit colloquial and might be difficult for non-native English speakers to follow (eg. potential for confusion with 'lumped' models).

L191-192: Please include a reference for this ('While shallow groundwater is expected to occur in regions with smaller watersheds...').

L197: Missing words: '...wide range of environments IN WHICH shallow groundwater can be present'.

L210: See point above regarding the physiographic provinces. As a non-American, I don't know where/what the 'Coastal Plain' province is. Please add some text in parentheses giving more detail.

L229: Missing word: 'We hypothesize THAT the lack...'

L232: Replace 'routes' with 'reroutes' or 'diverts'.

L375: Remove 'phase change', and just leave 'freeze-thaw'. At a first glance, this is slightly confusing as I thought it has something to do with the Tw/Ta phase lag.

L404: 'Less than -10' is a bit difficult to get my head around in this context. Maybe instead say 'a greater magnitude negative phase than -10'?

L406-408: It's not clear what was done with the 413 sites that had a negative lag between 0 and -10. Were these data thrown out, or were they used in the analyses?

L408: Are these references (12, 26) correct? Or should it be #11 and #24? Please check the references throughout this (methods) section, as I'm not sure that the reference numbering is always correct.

Reviewer #2 (Remarks to the Author):

The manuscript is an exciting potential contribution to our collective understanding of the role of groundwater inputs to sustaining freshwater streams and rivers at the continental scale. The manuscript uses a large dataset to explore spatial and temporal patterns, and seeks to identify factors that explain the observed patterns (e.g. human disturbance; geology). While I am excited about this manuscript, I believe the authors have an opportunity to improve their analysis and further the impact of this work through some additional analyses. While this means additional work, I strongly encourage the authors to consider addressing the questions below for this examination is unique and highly relevant to the sustainability of freshwater ecosystems, for which we all depend (water extraction for drinking water/irrigation for crops; industry; ecosystem services). There has also been a great emphasis on flooding within the larger community; the role of groundwater, especially during critical water stressed times, is a greater challenge as population increases and the climate warms.

Groundwater discharge ecosystems. Much of the literature with respect to GDEs is focussed on springs. Here, the authors have an opportunity with this dataset to describe the larger role of groundwater discharge across a large region (e.g., continental US) and explore not only shallow/deep but include other explanatory variables. For example, beyond the disturbance index, I'm very curious about changes in geology (e.g. depth to bedrock, soil characteristics) combined with hydroclimatology (e.g. snowmelt, rainfall, etc.) may help explain variation in the dataset. Furthermore, patterns wrt drainage area (or stream size) are also of much interest to the larger community and build on current work across large regions exploring water quantity and quality. For example, within the manuscript, the authors describe variation in shallow vs deep gw within the same basin - one has to ask if these changes are the result of GW recharge occurring within one portion of the basin that contribute to streamflow downstream. Here, the authors have an opportunity leverage their dataset and more deeply explore drainage size and other explanatory variables to explain how continental "spatial patterns and landscape drivers" alter groundwater discharge.

The authors also have an opportunity to provide insight into the role of deep vs shallow gw during lower flow periods (e.g. summer) when ET/human water demand is high, rainfall is lower, and stream ecosystems are most susceptible. While the BFI was incorporated at the annual scale, from an ecosystem and human water management point of view, late summer when streams exhibit their lowest flows will be of most interest. The authors are encouraged to incorporate a temporal component into their results which would provide insight into susceptible freshwaters.

Questions:

1. As I understand it, each classification is at the "annual" time scale. Is it possible to ask the question at shorter time scales - - e.g. spring, summer, fall, winter?
2. Beyond impervious surface area (paragraph line 222), groundwater is impacted significantly by over-allocation (and pumping). However, the focus of this paragraph is centered on impervious surface area (there is mention of withdrawal, but minimal, and emphasis seems to be on urban systems). Does the HDI incorporate water withdrawals or groundwater pumping that are likely a factor in altering gw discharge? Across the midwest and now the coastal plain of the southeastern US, groundwater extraction is increasing. Is this a partial explanatory variable, and if so, incorporate into the manuscript. As climate warms and gw dependence increases, the expectation is that gw discharge that sustains summertime streamflow will become less reliable and require a greater look at the combined surface and groundwater system. In many states across the US, freshwater management does not explicitly incorporate the coupled nature of surface and groundwater. This dataset has the potential to shed light on this importance across many regions in the US.
3. Line 294-295. While I am excited by the dataset and methods, I did not walk away with a take home message related to hydrogeological setting or watershed position. I feel as though this is a missed opportunity, and an analysis incorporating drainage area would increase the impact of your contribution.
4. The categorization relied on threshold values. Are these indeed representative? Is it possible to provide some sensitivity analysis to these 2 parameters? Given the range in physiographic provinces across the US, providing a more robust sensitivity analysis will bolster your findings.
5. This question was asked in the paragraph above, but for emphasis: What are the ratio of storm flow to groundwater-derived across regions? Is there a pattern with respect to drainage area? Can your analysis speak towards this in terms of shallow/deep GW inputs across the "annual" hydrographic?

Specific comments:

Line 25. What is meant by strong groundwater contribution? Can you quantify the amount of year fed by groundwater, compare to the annual volume of deep groundwater vs. shallow GW vs. runoff (storm related)?

Line 33. What is meant by connectivity here?

Line 38. Missing a verb.

Line 40. Provide references after drought conditions.

Line 40. Consider rephrasing groundwater discharge to groundwater inputs.

Line 40. Stream reaches - will the layperson know what a reach is?

Line 132. What is meant by "thus not normally distributed"?

Line 377. Methods - include a conceptual figure and date for a station. I would encourage you to include example stations for shallow/deep/atmospheric and the temperature trends.

Line 415. The selection of 10-days appears based on 2 studies, one in Cape Cod and one in Virginia. Are these indeed representative?

Terminology:

consider your more general audience to have the largest impact. Some terms within the abstract

that some readers may struggle with:

Groundwater dependent ecosystem - what is this? The next line of abstract (line 21) then uses the term stream ecosystem. For a select few, GDE's are at the outlets of springs which is not what you solely mean here.

Stream network - replace with watershed?

Flow path depth - replace with deep groundwater

Reviewer #3 (Remarks to the Author):

The manuscript "Continental-Scale Analysis of Shallow and Deep Groundwater Contributions to Streams" by Hare et al provides valuable insight into the spatial patterns, and to a degree temporal trends, of groundwater contributions to streams in the continental US. Such analysis is innovative and highlights the inter-connectedness of integrated groundwater and surface water resources. The authors base their analysis of groundwater discharge patterns on the comparison of time series in air and surface water temperatures, interpreting observed thermal regime signatures as result of variable shallow or deeper groundwater contributions. The analysis is scientifically robust and limitations in interpretations arising from uncertainties related to confounding impacts of other drivers of stream thermal regimes are discussed critically.

The study has potential to make a relevant contribution to the field and appeals to an interdisciplinary readership. I recommend publication in Nature Communications after the authors address a couple of points in their revision of the paper that I outline below:

I encourage the authors to clearer justify the discrimination between shallow and deeper groundwater and better explain their motivations for distinguishing between a shallower, less buffering and a deeper groundwater resource with more thermal buffering capacity. I am missing a discussion of the implications for the management and protection of different, shallow and deep, groundwater resources. I think the results provide ample opportunity to discuss in more depth potential consequences for the extraction (and mining) of deeper groundwater resources with consequences to not only the depletion of such resources themselves but their lost buffering impact on stream thermal regimes. What implications do the results of this study bear for groundwater management such as managed aquifer recharge, fracking and other mining activities?

Also, what are the implications beyond variable thermal buffering capacity of different groundwater resources? Could information on the depth of groundwater contributions to rivers be useful also for wider water quality management, with shallow groundwater resources being more likely to be impacted by diffuse nutrient pollution, pesticides, etc and hence, locations of shallow vs. deep groundwater upwelling indicate risks of different legacy pollution inputs into rivers?

With regards to the discriminatory criteria for deep vs. shallow groundwater – the mentioned 6 meters seem rather arbitrary and surely, will vary a lot based on the physical characteristics of the strata as well as water residence times. I would assume atmospheric thermal signatures to propagate much faster and further in dual porosity systems such as karst than in highly impermeable systems of course. The implications of this should be discussed explicitly. In fact, what is the point of having a depth criterion for discriminating between deep and shallow groundwater resources anyway – is this not all about responsiveness and thus, rather about faster and slower responding systems? It might help here to briefly discuss assumptions of the different mechanisms that govern groundwater temperature, from heat conduction to advective heat transport of vertical groundwater recharge. With that in mind – what relevance do the authors expect larger scale hyporheic exchange to have, in particular large-scale inter-meander flow?

Some more in-depth discussion of the purpose of the trend analysis as well as the limitations arising from a rather short time series is required. How much of a trend can you realistically

expect from 14-30 years of data that are confounded by so many different factors?

With that regards, the authors briefly mention in the methods section that they are not aware of any published evidence that other factors such as channel confinement, shading etc could cause the observed Ar – It would add robustness to the approach though to provide some indicative numbers at least what range of impacts they would expect from, riparian vegetation, deep channel incision etc and what variability in buffering impact from those confounding factors they would expect between up-stream and down-stream sections of river networks. At a continental scale – how important are the contributions of deeper geothermal groundwaters that would lead to misinterpretations as shallow groundwater inputs or even absence of groundwater upwelling?

Please plot the colour dots in Figure 1b to a similar size as in Figure 1a – otherwise they are barely visible.

And finally, the manuscript requires a thorough check of grammar and orthography - there are a few misspellings and text repetitions that need to be cleaned up that I, in the absence of line numbers, did not outline in detail here.

I am confident the authors will not have much trouble addressing the comments above in their revision and am looking forward to hopefully seeing this manuscript being published in the near future.

Stefan Krause

REVIEWER COMMENTS

Reviewer #1 (Remarks to the Author):

Nature Communications manuscript NCOMMS-20-22175 describes a novel study to assess groundwater (GW) contributions to streams across the contiguous USA. The study is impressive in both its geographic scope and its key findings, namely a) that shallow groundwater driven streams may be more sensitive to climate change than previously suspected and b) that traditional baseflow separation approaches may not be capable of revealing finer-scale differences in groundwater contribution needed for effective management of temperature sensitive streams. I also appreciated the authors' thorough and interesting interpretation of geographic trends in GW contributions in relation to established hypotheses regarding factors driving deep vs. shallow groundwater inputs, and the section on stream temperature temporal trends is particularly important. I therefore see this manuscript as a potentially significant and original contribution to improved understanding and management of streams threatened by climate change, and feel that the results will be of broad appeal to river managers and researchers working at the intersection of climate change, river ecology and hydrology/hydrogeology. This being said, despite these largely positive comments, I do have some moderate suggestions/improvements (listed below) that I feel would further strengthen a revised version of the manuscript; these relate primarily to clarifying questions regarding their methodology and potential limitations of their approach. Provided they are able to account for these changes, I would support the manuscript's publication in Nature Communications.

We thank the reviewer for the support of our manuscript and the thoughtful comments that have substantially improved our revised manuscript and led us to do some additional analyses to evaluate the robustness of our methods. We include our detailed responses to each general and specific comment below.

General comments

(1) • *Many of the paired Tw/Ta sites that the authors have used in their analyses comprise less than 3-4 years' (recent) data; I think it is important to demonstrate that the use of these relatively short (and recent) time series have not biased the results. One simple way to do this would be to compare the stability of the amplitude/lag metrics from a handful of shorter time series to nearby stations with longer records to ensure that the shorter series are not biased. The use of short time series also has implications for baseflow separation analyses, as 5 years' data is often suggested as a minimum time series for a robust BFI calculation. It would therefore be good to incorporate some discussion regarding this and potential limitations.*

We agree that the length of the record has the potential to influence the results and is important to consider; therefore, in response to this comment we compared sites with 2,3,and 4 year datasets and found no significant differences between these record lengths for phase lag or amplitude ratio. In addition, Briggs et al. (2018a) calculated that the average standard deviation using a record length of 1-yr compared to the full record length (~3 to 4 yr) was only 0.02 for amplitude ratio and 1.3d for phase lag, further supporting our analyses use of 2-4 years. We would also like to note here that we use recent temperature records because we want our results to reflect the most current hydrologic conditions, and to reduce influence by previous landscape and watershed conditions.

The reviewer also makes a good point to ensure the baseflow separation methods are applied for appropriate and comparable time intervals. Therefore, where the discharge data were available, we expanded each time-period of analysis to 5 years, using the same end date as the previous analysis that directly coincides with the annual temperature signal processing. Graphical results are pasted in below; in summary there was only a mean of 1.6% difference from the 2-4 year time period BFI analysis, and no systematic difference with BFI magnitude was apparent (variation around 1:1 line below).

We also went through the documentation for the USGS Groundwater Toolbox that bundles several baseflow separation methods, and that text indicates a time series of at least 1 year should be applicable to all methods listed there, including the BFI technique we apply. That report also makes the general statement: "*the estimates are most reliable when averaged over longer time periods, such as years.*" This statement seems to have as much to do with including a range of natural hydrologic variability to limitations of the specific techniques. We then contacted two of the Groundwater Toolbox authors, Bill Cunningham and Paul Barlow, and both indicated through email communication that they were not aware of a 5-year requirement and felt the 2-4 years of data we utilized should work well for the BFI method. Thus, given the lack of substantial difference in results (Figures pasted below) and guidance from the USGS Groundwater Toolbox, we conclude that our original time periods are appropriate and retain those results in the manuscript to be most comparable to the temperature signal data, which is more limited in record length than available river discharge data, and

directly corresponds with the annual temperature data the categories presented are based on.

(2) • Another slight concern is the lack of validation regarding the attribution of deep or shallow GW contributions (for sites outside of the Quashnet and the Shenandoah). I appreciate that gathering isotope/geochemical/borehole data for a multitude of rivers is beyond the scope of this manuscript, and I therefore stress that I do not feel that this limitation is a major issue. However, it would nonetheless be useful to present some information regarding how well the findings (in terms of deep vs. shallow GW contributions) conform to previously established groundwater contribution data (from isotopes separation or similar) for other select rivers, if such data exist. If these data are not available, I think that at minimum it would be important to acknowledge this as a potential limitation.

The reviewer has identified a primary drawback in the development of high-level groundwater/surface water exchange methodology based on widely available data types such as temperature: supporting data at comparable scales is often patchy/lacking. The predominance of groundwater/surface water exchange-based studies have been conducted at the reach scale, and rich biogeochemical datasets that the reviewer points to are often only available for select sub-regions covered by LTER or SFA networks (or similar). We validated our approach by comparing to the baseflow index (BFI) where possible, as stream discharge is the most available overlapping dataset.

A total of 554 sites did have hydrograph data available for analogous periods as the temperature data, and we believe the inclusion of the BFI analysis strengthened the study substantially. The results follow theory in that in general sites with a lower BFI do not display a strong groundwater influence signature (deep or shallow), and shallow groundwater sites in general have a lower BFI than deep groundwater sites. We have added text to clarify that this is an approach to validate our analysis. Unfortunately, we are unaware of other types of groundwater discharge process-related data that are available at similar scales.

Line 177-180 “Specifically, we calculated the baseflow index (BFI), an estimate of the ratio of baseflow to total streamflow based on the annual stream hydrograph, as it is one of the few current methods for quantifying relative groundwater contributions to streamflow efficiently at broad scales³⁰.”

Certainly, a subset of the stream sites presented in this paper are likely to have a suite of supporting data such as geochemical, isotopic, and perhaps introduced tracers and seepage runs. But given this would only cover a small percentage of the overall sites, and these sites are not likely to be well distributed across geographical regions and climate, it was not clear how other types of data could reasonably be included except for ‘select’ sites as the reviewer notes, and that approach seemed too likely to result in a cherry picking of locations without statistical significance in relation to the total dataset. We do compare to widely available watershed physical characteristic data such as slope and impervious area, and the apparent relations observed between some physical controls and stream site groundwater signature classification type would seem to

support the validity of our analysis. We have modified the following text to clarify this within the main text.

Line 78-88: “Though understanding the implications of climate change and land development for stream ecosystems requires quantifying the magnitude and source-depth of groundwater discharge, we lack efficient and broadly applicable methods to characterize source groundwater depth. Most hydrologic techniques for evaluating the physical properties of groundwater discharge are labor-intensive and not spatially and temporally scalable²⁸. More efficient methods, such as stream water temperature sensitivity linear regression analyses²⁹ or physically-based hydrograph separation techniques³⁰ do not directly differentiate groundwater source-depth. Inference regarding groundwater source-depth is possible using water chemistry end-member mixing³¹ or water isotopic data³², but these analyses cannot inherently specify shallow groundwater flow paths without additional supporting data and hydrologic characterization, and are time and resource-intensive.”

Given that we already highlight our groundwater signature classification from the North Fork Clear Water - Lake Creek watershed in Idaho in Figure 1b we did scour the literature for supporting analysis/data related groundwater connectivity in this area. A few state fisheries report this watershed is known to be groundwater dominated (nice verification of our results) and important habitat for a range of trout, but several of the headwater streams now exceed thermal tolerance criteria. One stream in particular (Lake Creek) was moved to a list of 'impaired' waters for summer warming in 2010; our analysis of the study site along this stream and within the contributing watershed indicate a dominance of shallow groundwater. We have now added the following text to the manuscript:

Line 255-264: “Interestingly, a major tributary (Lake Creek, highlighted in Figure 1b) was moved to the list of impaired waters in 2010 by the Idaho Department of Environmental Quality for elevated temperature criteria violations⁴⁷. Without explicit consideration of groundwater dynamics, this impairment was attributed to a slight reduction in canopy shading (4%) compared to the local shade optimal target. However, of the four sites we investigated in upper Lake Creek watershed, one main stem and three tributaries, all are classified as having shallow groundwater signatures of greater than 15-day phase lags. These large phase lags suggest dominance of the annual thermal regime by shallow groundwater, and we speculate that the previously observed warm stream impairment is due in part due to warming of shallow groundwater.”

• I liked the discussion regarding the heterogeneity of GW contributions within individual watersheds, but I do wonder to what extent differences in phase lag and amplitude between upstream and downstream locations within a same watershed could actually be explained by variability in energy fluxes at the air-water interface and the advection of heat from upstream, rather than due to local changes in GW contribution. For example, could the transition from open terrain to riparian tree cover (and the change in insolation that this entails) be responsible for a similar type of change in the T_a/T_w amplitude signal to that of a change in GW contribution? It would be good to include some discussion regarding this point.

The Reviewer touches on an important point, that being the annual temperature signal at any point in the network is influenced by a range of upstream heat exchange processes via advection. A stream water 'equilibrium temperature' is reached when all heat fluxes along the river corridor (positive and negative) are in balance. In typical headwater streams the true equilibrium temperature is rarely achieved due to major changes in factors such as elevation and land cover over short distances, along with perturbations resulting from spatially variable groundwater discharges. As referenced in the main text the annual signal method was developed in part using data from 120 sites across 18 watersheds in the headwater systems of Shenandoah National Park (Briggs et al 2018a,b) and has since been applied regionally to Pacific Northwest subwatersheds (Johnson et al 2020); we see no systematic changes in annual paired air/water signal metrics with downstream distance in either of these higher spatial resolution datasets. This indicates a fundamental control on these signal metrics is spatially heterogeneous deep/shallow groundwater connectivity. Additionally, some of the co-authors here are part of a manuscript under review that systematically addresses GW versus heat flux influences on annual stream temperatures. That study showed that GW fundamentally changes annual stream temperature signals differently than heat flux processes.

Certainly a transition from open canopy to heavy shading may serve to reduce or stabilize stream annual signal amplitude with downstream distance. Ideally the annual paired air/water signal method would be applied using local streamside air temperature data, which in this scenario would also have reduced annual signal amplitude, and the shading effect would then be accounted for when calculating the water/air signal amplitude ratio. However, for this continental study streamside air temperature was often not available so we used "local" air temperature stations within 25 km of the stream site. This approach might be expected to then result in spatially variable water/air amplitude ratio from differences in solar radiation alone (e.g. shading). To help account for this we have chosen a conservative amplitude ratio of 0.65 to indicate groundwater inflow that we do not expect to result from heavy shading alone. Further, shading is not expected to induce a multi-day air to water signal phase lag (especially >10 day, which is the threshold applied here) an effect that is believed to be unique to strong, shallow groundwater discharge or managed-flow, such as dam operations. The example watershed data we show in Figure 1b exemplifies strong phase lags throughout the steep headwaters that dissipate downstream, while downstream temperatures have a <0.65 amplitude ratio indicating deeper groundwater damping of stream signals, until the outlet that shows an atmospheric-dominated thermal regime. We chose this example specifically to not risk overinterpreting spatially-variable amplitude ratio which is likely driven in some capacity by differential shading, as the Reviewer notes, and instead better exemplifies a likely transition from shallow to deep groundwater discharge influence which generally follows conceptual theory. The methods text has been modified to state the following:

Line 518-521: "The hydrogeology of the Quashnet River has been extensively characterized^{64,65}, indicating streamflow is dominated by deep groundwater discharge that at times makes up close to 100% of total streamflow. Using a dynamic sinusoidal regression technique, Briggs et al⁸

found that A_r ranged from approximately 0.49 to 0.63 over a 3-year period with varied climatic conditions. Thus, we chose a threshold of 0.65 to indicate a deep groundwater signature for our study.”

• The decisions regarding the use of the A_r and $\Delta\phi$ values for classification as shallow vs deep groundwater are nicely justified in the text, but I would be interested to know what the results look like if these thresholds were relaxed somewhat. Have the authors done any sensitivity testing in this regard? Also, is there a case to be made for illustrating rivers along a ‘gradient’ of deep to shallow groundwater contribution (similar to Johnson et al. 2020, STOTEN), rather than using hard classification thresholds?

The choice of static thresholds for this analysis is an important point to highlight and we agree that describing groundwater influence along a gradient is more in-line with reality. We have added a figure to the Supplemental Materials to demonstrate the data distribution and placement of thresholds. However, for a national-scale study we do not believe it is currently possible to develop such a gradient with any confidence due to regional variation in the major components of stream heat budgets. As discussed in the text, we believe the thresholds chosen for this national scale data set for air-water phase lag (>10d) and water/air amplitude ratio (0.65) are conservative, and therefore likely indicators of deep and shallow groundwater influence (Line 509-538). Also, as noted in the previous comment, streamside air temperature records are not available and is thus another reason that thresholds are appropriate.

In reality most streams in the continental USA will exhibit some gradient of groundwater influence, as shown in part by the wide range of baseflow indices presented in this paper. We believe that regional temperature-signal based gradient scores of deep and shallow groundwater signatures can reasonably be developed at the regional level, as we previously done for Shenandoah National Park by Snyder et al (2015) and Johnson et al. (2017) (both cited in the manuscript) and mapped here using shorter term (sub-annual) air/water temperature data:

https://chesapeake.usgs.gov/shenandoah_groundwater/. We have modified the text to include this sentiment, which is shown below:

Line 535 - 538 “ A_r and $\Delta\phi$ thresholds may vary among watersheds and regions and thus can and should be modified based on additional information about individual watersheds for more precise, localized analyses. However, for the purposes of our analysis these thresholds represent conservative estimates applied across broad spatial scales.”

Further, in response to this review comment we explored the distribution of paired air/water temperature signal metrics in more detail with respect to our chosen ‘hard’ threshold classification system. The Supplementary Figure 1 shows the proportion of sites near the phase lag and amplitude ratio thresholds. In general, a low proportion (< ~4%) of sites are within 1 d of the phase lag threshold (10 d) and within 0.025 of the amplitude ratio threshold (0.65). We also conducted a Jenks natural breaks optimization method which identified an amplitude ratio of 0.72 and a phase lag of 7 days, as the

natural breaks within our analysed population. The following text was added to present these additional data and analyses.

Line 168-173: “Deep and shallow groundwater contributions to streamflow are not mutually exclusive, often a spectrum of flow path depths contributes to streamflow⁴⁰, but our analysis derives which signature is dominant. The distribution of annual signal metrics within our groundwater contribution categories indicate that our thresholds that define the groundwater signature categories occur near natural breaks (Supplementary Fig. 1), indicating alignment with potential groundwater-driven separations of underlying populations in the data”

and

Line 522 – 524: “These values were also supported by the natural breaks within our data populations ($A_r = 0.72$ and $\Delta\phi = 7$ days), based on Jenks' natural break method⁶⁶.”

• The 'Implications' section is good but it seems overly long and a little repetitive. I therefore feel that it could be cut down by about 1/3. Please go through this section and remove any detail that has already been covered in the preceding text.

Based on this comment and comments from other reviewers (see additional responses below), we have thoroughly revised the Implications section (now Conclusions section). In response to this comment, our revision includes removing or streamlining statements that focused on summarizing our results (and thus were repetitive of prior text), including reducing the length of the section from four to three paragraphs, which resulted in a ~100 words removed.

Specific comments

L38: Missing word: ‘...zones ARE an important driver...’ This sentence has been modified

L50: I’m not sure if this clause (‘Earth’s most recent geological time period...’) is necessary. This text has been removed.

L55: Suggest including a hyphen in ‘source-flow path characteristics’ This change has been made throughout.

L63: Move reference 14 to end of this sentence (‘...low permeability bedrock’). This text has been removed.

L102: Please include a one-sentence explanation of what a ‘physiographic province’ is for non-American readers? This sentence was modified to: ‘...across 21 of the 25 U.S. physiographic provinces (categorized based on large scale geomorphology)’.

L109-112: It would be helpful to spell out why shallow groundwater contributions shift the phase lag of the Ta/Tw signal. The following text was added: ‘This characteristic phase lag propagates into stream water from adjacent shallow aquifers, whereas deeper groundwater flow paths have a highly attenuated annual temperature signal thus the phase of groundwater does not influence the surface water signal phase.’ Line 137- 140.

L129-130: This sentence (‘Within a stream...’) seems to be missing something. Please rewrite for clarity. This sentence was revised.

L132: Put ‘thus not normally distributed’ in parentheses. This sentence was revised.

Figure 1: Please consider colouring the actual rivers/river reaches rather than the individual points. I know that technically speaking, the computed GW contribution class only applies to the exact point location where the Tw logger is installed, but I think that it would make the figure much more visually appealing if the rivers themselves were coloured.

This is an interesting point. However, the upstream spatial extent to which each stream site annual signal metric categorization applies can be highly variable as shown by high spatial resolution regional analysis of Briggs et al (2018b) and Johnson et al (2020). Therefore, we decided to stick with just marking the exact points of the stream temperature sites to avoid overly subjective decisions regarding what the upstream extent of the analysis might reflect. The BFI calculation has a similar limitation though may be expected to better represent the upstream contributing area more holistically as hydrological pulses are potentially more conservatively propagated downstream (except in losing systems) compared to heat.

L164: Remove 'annual temperature signal based' from this sentence – it makes it difficult to follow. This change has been made.

L167: Missing word: 'We hypothesize THAT this difference...' This change has been made.

L172: Please rephrase this sentence. The word 'lumping' is a bit colloquial and might be difficult for non-native English speakers to follow (eg. potential for confusion with 'lumped' models). This sentence was revised.

L191-192: Please include a reference for this ('While shallow groundwater is expected to occur in regions with smaller watersheds...'). This sentence was revised to read "Yet, our results show that strong connectivity of streams and shallow groundwater occurs in environments beyond smaller, steep headwater streams, such as areas with shallow confining layers."

L197: Missing words: '...wide range of environments IN WHICH shallow groundwater can be present'. This sentence was revised.

L210: See point above regarding the physiographic provinces. As a non-American, I don't know where/what the 'Coastal Plain' province is. Please add some text in parentheses giving more detail. "(eastern coastline of the United States from Massachusetts to Mexico)" was added here.

L229: Missing word: 'We hypothesize THAT the lack...' This change has been made

L232: Replace 'routes' with 'reroutes' or 'diverts'. This change has been made to 'diverts'.

L375: Remove 'phase change', and just leave 'freeze-thaw'. At a first glance, this is slightly

confusing as I thought it has something to do with the T_w/T_a phase lag. This change has been made.

L404: 'Less than -10' is a bit difficult to get my head around in this context. Maybe instead say 'a greater magnitude negative phase than -10'? This sentence was revised to 'Negative phase lags greater than 10 days'

L406-408: It's not clear what was done with the 413 sites that had a negative lag between 0 and -10. Were these data thrown out, or were they used in the analyses? Text has been added to clarify that these sites were kept within the analysis.

L408: Are these references (12, 26) correct? Or should it be #11 and #24? Please check the references throughout this (methods) section, as I'm not sure that the reference numbering is always correct. These citations have been corrected.

Reviewer #2 (Remarks to the Author):

The manuscript is an exciting potential contribution to our collective understanding of the role of groundwater inputs to sustaining freshwater streams and rivers at the continental scale. The manuscript uses a large dataset to explore spatial and temporal patterns and seeks to identify

factors that explain the observed patterns (e.g. human disturbance; geology). While I am excited about this manuscript, I believe the authors have an opportunity to improve their analysis and further the impact of this work through some additional analyses. While this means additional work, I strongly encourage the authors to consider addressing the questions below for this examination is unique and highly relevant to the sustainability of freshwater ecosystems, for which we all depend (water extraction for drinking water/irrigation for crops; industry; ecosystem services). There has also been a great emphasis on flooding within the larger community; the role of groundwater, especially during critical water stressed times, is a greater challenge as population increases and the climate warms.

We thank the reviewer for the support of our manuscript and for asking questions that broadened or refined our consideration of landscape and seasonal patterns in our analysis, greatly improving the revised version. We respond to each comment in detail below:

*Groundwater discharge ecosystems. Much of the literature with respect to GDEs is focused on springs. Here, the authors have an opportunity with this dataset to describe the larger role of groundwater discharge across a large region (e.g., continental US) and explore not only shallow/deep but include other explanatory variables. For example, beyond the disturbance index, I'm very curious about changes in geology (e.g. **depth to bedrock**, soil characteristics) combined with hydroclimatology (e.g. snowmelt, rainfall, etc.) may help explain variation in the dataset. Furthermore, patterns wrt drainage area (or stream size) are also of much interest to the larger community and build on current work across large regions exploring water quantity and quality. For example, within the manuscript, the authors describe variation in shallow vs deep gw within the same basin - one has to ask if these changes are the result of GW recharge occurring within one portion of the basin that contribute to streamflow downstream. Here, the authors have an opportunity leverage their dataset and more deeply explore drainage size and other explanatory variables to explain how continental "spatial patterns and landscape drivers" alter groundwater discharge.*

We agree that there are many variables (that we are also curious and excited about) that may explain additional variation in our datasets, yet we also recognize that we are constrained by the length of a Nature Communications article and the availability of datasets at a quality, spatial coverage, and resolution appropriate for our analysis. In response to this comment, we have expanded and emphasized our analysis of watershed size in the revised manuscript. We also have added a comparison to average stream discharge data, as often the contributing groundwater basin is larger than the contributing watershed, thus may elucidate varying aquifer dynamics that are not captured in the watershed size.

Line 242-248 "Sites with shallow groundwater signatures drain larger watersheds (median - 153 km² ; Q1-Q3: [17 km² , 2131 km²]), have higher streamflow (median - 444 cfs), and have a greater range of streamflow (Q1-Q3: [62 cfs , 3466 cfs]) than sites with deep groundwater signatures (watershed size: 65 km²; Q1-Q3: [18 km² , 616 km²]; streamflow 67 cfs; [13 cfs, 338 cfs]) suggesting shallow groundwater signatures occur across a wide spectrum of hydrogeologic settings that may not be predicted by current conceptual models of baseflow generation."

We considered including depth to bedrock, particularly since Briggs et al (2018b) shows depth to bedrock is likely a primary driver of shallow groundwater discharge in headwater stream networks. However, the only continental scale data we are aware of is the United States Department of Agriculture's soil maps, in which "depth of bedrock" within this dataset is the true depth of bedrock or 2m, whichever is lesser. Since our delineation of shallow versus deep groundwater is at approximately 6 m, this dataset is not informative for our study.

We also carefully considered hydroclimatology, which we agree could provide insight into regional patterns. However, an appropriate analysis would include aligning the dates of our analysis with regional rainfall and/or snowfall, which would require significant code development, and would be beyond the scope of this manuscript. We feel this would be an appropriate and sizeable analysis for an additional stand-alone paper.

Finally, our current analysis relies on linking our annual temperature signal data to StreamCAT and GAGES-II data, and many of the variables were explored (altitude, average streamflow, elevation of station, mean/median elevation of watershed, population density, housing density, bedrock geology), but we only presented the results that were strongly relevant for the scale of study and not intercorrelated. We encourage future research into regional studies where more variables can be compared at appropriate scales.

The authors also have an opportunity to provide insight into the role of deep vs shallow gw during lower flow periods (e.g. summer) when ET/human water demand is high, rainfall is lower, and stream ecosystems are most susceptible. While the BFI was incorporated at the annual scale, from an ecosystem and human water management point of view, late summer when streams exhibit their lowest flows will be of most interest. The authors are encouraged to incorporate a temporal component into their results which would provide insight into susceptible freshwaters.

We agree, as summer conditions are of particular concern for many ecosystems, we have added in an analysis of summer stream temperature trends, both within the text and an additional figure panel (now Figure 4c), to strengthen the discussion as to the importance of shallow versus deep groundwater contributions during individual seasonal conditions. The fundamental nature of the annual signal temperature analysis is that this method relies on the pronounced, highly sensitive, annual temperature signals of shallow groundwater compared to the buffered annual temperature signal of deep groundwater. Therefore, sites with shallow groundwater signature will be susceptible to episodic, acute land surface temperature perturbations, while deep groundwater is more resistant to land surface temperature changes, but still responsive to longer-term, chronic thermal land surface changes. We have now added text to include this additional analysis.

Line 341-356 “The disparity between long-term stream temperature trends of sites with shallow versus deep groundwater signatures is also observed when comparing seasonal summer stream temperatures, when cold water fishes are most often thermally stressed. Over 70% of sites with shallow groundwater signatures show significant summer season warming trends compared to 43% of sites with deep groundwater and 61% of sites with atmospheric signatures (Figure 4c). These seasonal warming trends follow the fundamental nature of the classification method, which relies on the pronounced annual temperature signals of shallow groundwater to be transferred to stream water via groundwater discharge zones. Sites with shallow groundwater signatures will be immediately sensitive to hotter summers, exacerbating thermal stress on sensitive aquatic organisms³⁹. Deep groundwater is more resistant to land surface temperature changes, but still sensitive to longer-term thermal shifts at timescales tied to source flow path depth³⁶. These results indicate that vulnerable biota within streams dominated by shallow groundwater may not only have to adapt to a warming baseline condition, but also be exposed to single season land surface temperature spikes. This re-emphasizes the importance of distinguishing shallow versus deep groundwater source-depth, rather than assuming streams with strong baseflow components imply thermal stability.”

Questions:

1. As I understand it, each classification is at the “annual” time scale. Is it possible to ask the question at shorter time scales - - e.g. spring, summer, fall, winter?

The short answer is ‘yes’, paired air and water annual temperature signals are our fundamental ‘tracer’ of deep and shallow groundwater signatures in stream water, but the signal processing can be conducted in various ways to explore sub-annual groundwater discharge dynamics. For example, Briggs et al 2018a used a technique called ‘dynamic harmonic regression’ to extract non-stationary sine curve parameters (amplitude and phase) and compared to daily streamflow data. Johnson et al 2020 used sliding windows of analysis to track temporal changes in annual sine curve parameters. However, we did not see room in this national scale study to explore subannual dynamics that can be highly complex, and were instead more interested in developing a simple framework of classification using stationary sine curves applied to multiple years of daily data, in effect extracting the ‘average’ paired air/water thermal behavior of the various stream systems. We believe that this high-level classification is perhaps most useful, upon which more specific and in depth local to regional analysis can be conducted. Further, the use of multi-year sine curve fits reduces the potential for other short-term drivers of stream signal phase lag (such as snow melt pulses) to impact our high-level classification. We did however, look at how the classification differed temporally through the seasons and included summer trend analysis, as described above.

2. Beyond impervious surface area (paragraph line 222), groundwater is impacted significantly by over-allocation (and pumping). However, the focus of this paragraph is centered on impervious surface area (there is mention of withdrawal, but minimal, and emphasis seems to be on urban systems). Does the HDI incorporate water withdrawals or groundwater pumping that are likely a factor in altering gw discharge? Across the midwest and now the coastal plain of the southeastern US, groundwater extraction is increasing. Is this a partial explanatory variable, and if so, incorporate into the manuscript. As climate warms and gw dependence increases, the expectation is that gw discharge that sustains summertime streamflow will become less reliable

and require a greater look at the combined surface and groundwater system. In many states across the US, freshwater management does not explicitly incorporate the coupled nature of surface and groundwater. This dataset has the potential to shed light on this importance across many regions in the US.

The reviewer does raise an important point when considering streamflow dynamics, particularly when looking at long-term trends. We have now better emphasized the importance of groundwater withdrawal to human-modifications of stream flow within the text. Yes, the HDI does include a parameter of estimated freshwater withdrawal; however, we do not feel that we can separate these results from the metric as a whole in a meaningful way for distinct analysis.

Line 291-299 “One of these seven HDI parameters is groundwater withdrawal, which has been shown to have immediate effects on streamflow generation, especially within areas reliant on irrigation, and is generally projected to increase in the future to offset droughts⁵². We hypothesize that in addition to pumping, the lack of sites with groundwater signatures observed in this study in more disturbed landscapes is a result of the many human landscape modifications that reduce groundwater discharge to streams and rivers. These impacts occur either directly through groundwater withdrawal or indirectly through impervious cover and stormwater infrastructure that saps shallow groundwater and diverts precipitation to streams, reducing infiltration and aquifer recharge.”

3. Line 294-295. While I am excited by the dataset and methods, I did not walk away with a take home message related to hydrogeological setting or watershed position. I feel as though this is a missed opportunity, and an analysis incorporating drainage area would increase the impact of your contribution.

We have revised the sections ‘Spatial Patterns and Physical Drivers’ and ‘Conclusions’ to better emphasize our take-home messages related to hydrogeologic setting and watershed characteristics. We find that streams with deep groundwater signatures tend to occur more frequently in regions with productive aquifers and in watersheds with lower slopes while streams with shallow groundwater signatures tend to occur in regions with steep mountainous terrain with generally thin soil coverage and watersheds with higher slopes. Interestingly though, atmospheric, shallow, and deep groundwater signatures co-occur within all eight physiographic regions we considered in this study. In the manuscript, we posit that human alterations to watersheds could account for some of this heterogeneity since sites with groundwater signatures tend to have low impervious cover and human disturbance index.

Our original manuscript did include an analysis of drainage area, but it was not emphasized in the text; thus, we have added additional detail explaining patterns of drainage areas and streamflow among our categories and is described in our earlier response.

4. The categorization relied on threshold values. Are these indeed representative? Is it possible to provide some sensitivity analysis to these 2 parameters? Given the range in physiographic provinces across the US, providing a more robust sensitivity analysis will bolster your findings.

Based on this comment and comments from other reviewers, we have added a Supplemental Figure 1 that depicts the distribution of our amplitude and phase lag values with respect to corresponding thresholds. These results support the modeling and regional scale analyses used to generate the thresholds, as they correspond with natural breaks within the data, thus are appropriate given the national scale of this analysis. As the reviewer points out due to the wide range in physiographic provinces a full depiction of the data is useful, and for the same reasons (wide range of environments) thresholds are the most appropriate way to describe these data.

Reviewer #1 had a similar questions, and we feel that our response addresses this comment as well.

“The choice of static thresholds for this analysis is an important point to highlight and we agree that describing groundwater influence along a gradient is more in-line with reality. We have added a figure to the Supplemental Materials to demonstrate the data distribution and placement of thresholds. However, for a national-scale study we do not believe it is currently possible to develop a gradient with any confidence due to regional variation in the major components of stream heat budgets. As discussed in the text, we believe the thresholds chosen for this national scale data set for air-water phase lag (>10d) and water/air amplitude ratio (0.65) are conservative, and therefore likely indicators of deep and shallow groundwater influence (Line 509-538). Also, as noted in the previous comment, streamside air temperature records are not available and is thus another reason that thresholds are appropriate.

In reality most streams in the continental USA will exhibit some gradient of groundwater influence, as shown in part by the wide range of baseflow indices presented in this paper. We believe that regional temperature-signal based gradient scores of deep and shallow groundwater signatures can reasonably be developed at the regional level, as we previously done for Shenandoah National Park by Snyder et al (2015) and Johnson et al. (2017) (both cited in the manuscript) and mapped here using shorter term (sub-annual) air/water temperature data:

https://chesapeake.usgs.gov/shenandoah_groundwater/. We have modified the text to include this sentiment, which is shown below:

Line 535 - 538 “ A_r and $\Delta\phi$ thresholds may vary among watersheds and regions and thus can and should be modified based on additional information about individual watersheds for more precise, localized analyses. However, for the purposes of our analysis these thresholds represent conservative estimates applied across broad spatial scales.”

Further, in response to this review comment we explored the distribution of paired air/water temperature signal metrics in more detail with respect to our chosen ‘hard’

threshold classification system. The Supplementary Figure 1 shows the proportion of sites near the phase lag and amplitude ratio thresholds. In general, a low proportion (< ~4%) of sites are within 1 d of the phase lag threshold (10 d) and within 0.025 of the amplitude ratio threshold (0.65). We also conducted a Jenks natural breaks optimization method which identified an amplitude ratio of 0.72 and a phase lag of 7 days, as the natural breaks within our analysed population. The following text was added to present these additional data and analyses.

Line 168-173: “Deep and shallow groundwater contributions to streamflow are not mutually exclusive, often a spectrum of flow path depths contributes to streamflow⁴⁰, but our analysis derives which signature is dominant. The distribution of annual signal metrics within our groundwater contribution categories indicate that our thresholds that define the groundwater signature categories occur near natural breaks (Supplementary Fig. 1), indicating alignment with potential groundwater-driven separations of underlying populations in the data”

and

Line 522 – 524: “These values were also supported by the natural breaks within our data populations ($A_r = 0.72$ and $\Delta\phi = 7$ days), based on Jenks' natural break method⁶⁶.”

5. This question was asked in the paragraph above, but for emphasis: What are the ratio of storm flow to groundwater-derived across regions? Is there a pattern with respect to drainage area? Can your analysis speak towards this in terms of shallow/deep GW inputs across the “annual” hydrographic?

Essentially the definition of baseflow separation (BFI method used here) is the discrimination of storm flow effects on total streamflow from longer residence time streamflow sources that include soil drainage and groundwater discharge. We present the BFI as baseflow to total discharge, essentially the inverse of the ratio the reviewer mentions in this question. Therefore, we compared baseflow scores to watershed area where that data were available and found that larger watershed tended to have higher BFI, however, the correlation between these is low ($r^2 = 0.16$). In regard to analyzing these data on an annual time scale, we made the decision to evaluate the BFI over the same period as the temperature data to compare the same hydrologic conditions. We feel that analyzing the annual hydrographs to support our data may be misleading.

Specific comments:

Line 25. What is meant by strong groundwater contribution? Can you quantify the amount of year fed by groundwater, compare to the annual volume of deep groundwater vs. shallow GW vs. runoff (storm related)?

This sentence was modified to: ‘Approximately 40% of non-dam stream sites have substantial groundwater contributions. Sites with shallow groundwater signatures account for half of all groundwater signature sites.’

As applied here, the use of paired annual air/stream temperature signals is a qualitative metric of shallow and deep groundwater influence. The method can be applied more quantitatively as shown by Briggs et al 2018b, but that application necessitates the development of regional to local aquifer heat transport models that predict depth-specific annual temperature signal amplitude and phase, which are beyond the scope of this high-level national scale classification. The storm-related streamflow fraction is indicated quantitatively by the BFI score where flow data were available.

Line 33. What is meant by connectivity here? This phrase was modified to: ‘stream-groundwater connectivity’

Line 38. Missing a verb. This sentence was revised

Line 40. Provide references after drought conditions. Citation added: Kløve, B. *et al.* Climate change impacts on groundwater and dependent ecosystems. *J. Hydrol.* **518**, 250–266 (2014).

Line 40. Consider rephrasing groundwater discharge to groundwater inputs. We reviewed the river corridor literature at large, and although 'inputs' and 'inflows' are used extensively we have found 'discharge' to be most universal so decided to stick with that verbiage.

Line 40. Stream reaches - will the layperson know what a reach is? Perhaps not.... This sentence was modified

Line 132. What is meant by "thus not normally distributed"? This phrase was removed as the concept is now shown by the new supplemental plots.

Line 377. Methods - include a conceptual figure and date for a station. I would encourage you to include example stations for shallow/deep/atmospheric and the temperature trends.

Although we agree that a conceptual figure could be quite useful, the citation in this section to Briggs et al 2018a,b direct the reader to the detailed method development including multiple conceptual figures. The Supplemental material lists each station and the applied category (shallow/deep/atmospheric, ect).

Line 415. The selection of 10-days appears based on 2 studies, one in Cape Cod and one in Virginia. Are these indeed representative?

We believe 10 days to be a conservative threshold that is likely to indicate the signature of shallow groundwater, based on the data and theory presented in Briggs et al 2018b and the data presented in Johnson et al 2020. Given the wide range of hydrogeologic conditions nationally, the use of one static threshold is by nature somewhat imprecise, and there are likely stream sites with shorter phase lags that are also shallow groundwater dominated. The sensitivity of our categorizations to the threshold value choices is now more transparent in the new Supplemental Material.

Terminology:

consider your more general audience to have the largest impact. Some terms within the abstract that some readers may struggle with:

Groundwater dependent ecosystem - what is this? The next line of abstract (line 21) then uses the term stream ecosystem. For a select few, GDE's are at the outlets of springs which is not what you solely mean here. The GDE terminology seems to be growing in popularity based on our observations across scientific and more general literature/media so we thought it important and appropriate to include here. To support better this we have now added the reference: *Rivers as groundwater-dependent ecosystems: a review of degrees of dependency, riverine processes and management implications*, which explains GDEs occur much more broadly than in spring run creeks.

Stream network - replace with watershed? change made

Flow path depth - replace with deep groundwater replaced with 'source-depth'

Reviewer #3 (Remarks to the Author):

The manuscript “Continental-Scale Analysis of Shallow and Deep Groundwater Contributions to Streams” by Hare et al provides valuable insight into the spatial patterns, and to a degree temporal trends, of groundwater contributions to streams in the continental US. Such analysis is innovative and highlights the inter-connectedness of integrated groundwater and surface water resources. The authors base their analysis of groundwater discharge patterns on the comparison of time series in air and surface water temperatures, interpreting observed thermal regime signatures as result of variable shallow or deeper groundwater contributions. The analysis is scientifically robust and limitations in interpretations arising from uncertainties related to confounding impacts of other drivers of stream thermal regimes are discussed critically.

The study has potential to make a relevant contribution to the field and appeals to an interdisciplinary readership. I recommend publication in Nature Communications after the authors address a couple of points in their revision of the paper that I outline below:

Thank you for taking on this review and for pointing out several themes that could add *strength* to the *manuscript*.

I encourage the authors to clearer justify the discrimination between shallow and deeper groundwater and better explain their motivations for distinguishing between a shallower, less buffering and a deeper groundwater resource with more thermal buffering capacity. I am missing a discussion of the implications for the management and protection of different, shallow and deep, groundwater resources. I think the results provide ample opportunity to discuss in more depth potential consequences for the extraction (and mining) of deeper groundwater resources with consequences to not only the depletion of such resources themselves but their lost buffering impact on stream thermal regimes. What implications do the results of this study bear for groundwater management such as managed aquifer recharge, fracking and other mining activities?

A growing body of recent research has demonstrated the multifaceted reasons why shallow/near surface/critical zone groundwater is fundamentally different than groundwater that originates from deeper flow paths, and these differences impact stream processes via groundwater discharge zones. A few high level examples of how shallow groundwater are distinct include close coupling to surface heat exchange, close coupling to short and longer term climate variation, close coupling to land use change and related nutrients/contaminants (including surface leakage for produced fracking waters), susceptibility to enhanced transpiration in a warming world, and diverse organic carbon sources and microbial communities compared to deeper groundwater. We have added citations and modified text to reflect these implications more thoroughly.

As the Reviewer notes, deep groundwater also has unique potential stressors such as groundwater mining and resource extraction, but we focus on this in terms of

vulnerabilities of shallow groundwater. Multiple citations and text have been added to discuss this important topic more thoroughly.

Line 291-299 “One of these seven HDI parameters is groundwater withdrawal, which has been shown to have immediate effects on streamflow generation, especially within areas reliant on irrigation, and is generally projected to increase in the future to offset droughts⁵². We hypothesize that in addition to pumping, the lack of sites with groundwater signatures observed in this study in more disturbed landscapes is a result of the many human landscape modifications that reduce groundwater discharge to streams and rivers. These impacts occur either directly through groundwater withdrawal or indirectly through impervious cover and stormwater infrastructure that saps shallow groundwater and diverts precipitation to streams, reducing infiltration and aquifer recharge.”

Line 335 – 340 “We recognize that there are confounding factors that influence long-term stream temperature, notably discharge variability. Therefore, streams fed by shallow groundwater could warm at a faster rate in part because of drought conditions or groundwater withdrawal (e.g. for irrigation) lowering groundwater levels, which disproportionately affects shallow groundwater²⁶. Thus, shallow baseflow-dominated streams previously thought to be more resistant to surface warming due to high groundwater contribution to streamflow may actually be at a higher risk of temperature increase and be less likely to function as ‘climate refugia’⁵⁷.”

We explain our motivations for distinguishing between shallow and deep groundwater contributions to streams in lines 54 - 76 (third paragraph of intro) - we have revised this paragraph to more clearly illustrate the distinction between shallow and deep groundwater systems especially in the context of stream thermal buffering, water quality & ecosystem function, and vulnerability to climate change.

Also, what are the implications beyond variable thermal buffering capacity of different groundwater resources? Could information on the depth of groundwater contributions to rivers be useful also for wider water quality management, with shallow groundwater resources being more likely to be impacted by diffuse nutrient pollution, pesticides, etc and hence, locations of shallow vs. deep groundwater upwelling indicate risks of different legacy pollution inputs into rivers?

This is an important point that is now better addressed in the manuscript, particularly by the third paragraph of the Introduction and the Conclusions section, and the addition of multiple citations that directly discuss emerging contaminants, agricultural contaminants, and fracking waste. We have also bolstered language throughout to reflect important implications. We do demonstrate the implications explicitly within our BFI comparison analysis, as we show that, in general, stream sites with shallow groundwater signatures have reduced baseflow compared to deep groundwater signature sites, directly indicating the impact of source flow path depth on streamflow regimes.

With regards to the discriminatory criteria for deep vs. shallow groundwater – the mentioned 6 meters seem rather arbitrary and surely, will vary a lot based on the physical characteristics of the strata as well as water residence times. I would assume atmospheric thermal signatures to propagate much faster and further in dual porosity systems such as karst than in highly impermeable systems of course. The implications of this should be discussed explicitly. In fact, what is the point of having a depth criterion for discriminating between deep and shallow groundwater resources anyway – is this not all about responsiveness and thus, rather about faster and slower responding systems? It might help here to briefly discuss assumptions of the different mechanisms that govern groundwater temperature, from heat conduction to advective heat transport of vertical groundwater recharge.

The mention of approximately 6 m as a general threshold from below which we do not expect groundwater discharge to induce substantial (>10 day) instream annual signal phase lags is supported by both first principles of vertical heat exchange and mixing, as well as published data. The Reviewer correctly notes that the vertical propagation of land surface thermal signals is tied to the near surface geology (specifically the thermal diffusivity); for example weathered limestone or till has a strongly reduced thermal diffusivity compared to quartz sand-rich sediments, so annual signals are more conservatively transported downward in the latter setting. Similarly, saturated sediments with higher porosity will show reduced heat exchange due to the high heat capacity of water compared to most sediment grains. Recharge will also transport heat vertically via advection, so watersheds with greater precipitation might be expected to show stronger propagation of surface thermal signals with aquifer depth. Briggs et al 2018b used measured sediment thermal diffusivities from an idealized mountain headwater system to develop numerical advection-conduction models (including recharge) to predict shallow groundwater annual temperature dynamics. Then conservative stream/groundwater discharge mixing models were used to predict the instream convolution of annual signals. Based on this study at least 25% of total streamflow would need to consist of 6 m (or shallower) groundwater to induce a 10 d+ phase lag from local air. This theoretical work is supported by a number of other published model based and shallow aquifer temperature data, such as that presented by Bundschuh et al 1993.

New text added to Line 134-139 in Introduction section:

‘Groundwater discharge from shallow flow paths causes variable stream temperature signal damping, but uniquely shifts the timing of the annual stream water temperature signal relative to the annual air temperature signal—quantified by the time-forward *phase lag*. This characteristic phase lag propagates into stream water from adjacent shallow aquifers, whereas deeper groundwater flow paths have a highly attenuated annual temperature signal and thus do not influence the stream water signal phase⁸.’

With that in mind – what relevance do the authors expect larger scale hyporheic exchange to have, in particular large-scale inter-meander flow?

As the Reviewer is acutely aware, the definition of 'hyporheic' flow is not standard across the hydrology and ecology communities and is often interpreted in a situational context. Hyporheic flow can be defined as containing a certain fraction of surface derived water and/or based on a subsurface residence timescale (from inception to return to channel). For the purposes of this work we do not distinguish between relatively long residence time hyporheic flow paths (such as large-scale cross meander bend flow) where annual signal phase lags might develop from shallow groundwater flow paths, as this distinction would necessitate additional tracers (ie chemical) that are not broadly available for these stream sites. Research by Helton et al 2012 (citation now included) and others has documented substantial shallow flow path phase lag across alluvial river meander bends along with notable mixing along these flow paths with floodplain groundwater.

Some more in-depth discussion of the purpose of the trend analysis as well as the limitations arising from a rather short time series is required. How much of a trend can you realistically expect from 14-30 years of data that are confounded by so many different factors?

We determined that this length of time was appropriate for this study based on the El Niño-Southern Oscillation (ENSO) period, which is three to seven years, thus the minimum length of record (14 years) would encapsulate at least two full cycles. We recognize that these time series are short when accounting for Pacific Decadal Oscillations; however, available stream temperature records earlier than 1980s are rare (Issak et al. 2012), and our results do not indicate a strong geographic distinction between sites located in western states versus the rest of the U.S.A.. Other factors influencing stream temperature trends, notably discharge variability, are encapsulated within the conceptualization we present, and many of the other important temperature influences such as changes to snowmelt are seasonal effects and thus are effectively removed through the use of monthly averages and deseasoning the trend. We have added in text describing these limitations, as we agree they should be made more explicit.

Line 335 – 340 “We recognize that there are confounding factors that influence long-term stream temperature, notably discharge variability. Therefore, streams fed by shallow groundwater could warm at a faster rate in part because of drought conditions or groundwater withdrawal (e.g. for irrigation) lowering groundwater levels, which disproportionately affects shallow groundwater²⁶. Thus, shallow baseflow-dominated streams previously thought to be more resistant to surface warming due to high groundwater contribution to streamflow may actually be at a higher risk of temperature increase and be less likely to function as ‘climate refugia’⁵⁷.”

554-560 “We analyzed a subset of our stream water temperature records for monotonic 14-year to 30-year trends (January 1990 - December 2019). This record length was chosen to account for the El Niño-Southern Oscillation (ENSO) period, which is three to seven years, thus the minimum length of record (14 years) would encapsulate at least two full cycles. We recognize that these time series are short when accounting for Pacific Decadal Oscillations; however, our results indicate there is not a distinction between sites located in the western United States and the rest of the sites.”

Line 570-572 “We used the monthly averages to reduce autocorrelation and the ‘deseason’ option of the function to account for potentially important seasonal temperature influences such as changes to snowmelt.”

With that regards, the authors briefly mention in the methods section that they are not aware of any published evidence that other factors such as channel confinement, shading etc could cause the observed A_r – It would add robustness to the approach though to provide some indicative numbers at least what range of impacts they would expect from, riparian vegetation, deep channel incision etc and what variability in buffering impact from those confounding factors they would expect between up-stream and down-stream sections of river networks. At a continental scale – how important are the contributions of deeper geothermal groundwaters that would lead to misinterpretations as shallow groundwater inputs or even absence of groundwater upwelling?

Currently, some of the co-authors here are part of a manuscript under review that systematically addresses groundwater versus heat flux influences on annual stream temperatures using the metrics that the Reviewer has brought up. While we haven't/ cannot use that analysis directly within this analysis, we have been cognizant of the results and the influence of these factors. Notably, these factors, such as shading can influence the stream distance over which the contributing groundwater signature can be detected, but shading and other confounding factors are unlikely to induce the 0.65 amplitude ratio, nor the 10 day phase lag threshold we use to determine groundwater contribution.

Line 522-524 “Other physical factors such as channel confinement, aspect, and shading could affect A_r , but to date no published work that we are aware of indicates these factors could explain $A_r < 0.65$ without the influence of groundwater. However, we hypothesize that these factors are likely to change the distance downstream these annual signal signals can be detected.”

In regard to contributions of deeper geothermal groundwaters, our amplitude ratio relies solely on the amplitude of the air and surface water signals, not the magnitude value; therefore, a stream with geothermal groundwater contribution would be indicated as a deep groundwater signature due to the buffered annual signal, which would be an appropriate assessment in terms of groundwater connectivity. A high mean temperature ratio (surface water parameter C : air parameter C from equation 1) could indicate geothermal influence, but that was not incorporated in this analysis, as we focus on discriminates between near surface and all deeper groundwater flow paths. Also stream temperature values greater than 60C were removed from the analysis before data gaps were assessed.

Please plot the colour dots in Figure 1b to a similar size as in Figure 1a – otherwise they are barely visible.

This figure has been updated to reflect this comment.

And finally, the manuscript requires a thorough check of grammar and orthography - there are a few misspellings and text repetitions that need to be cleaned up that I, in the absence of line numbers, did not outline in detail here.

The current submission has addressed grammatical and structural edits.

I am confident the authors will not have much trouble addressing the comments above in their revision and am looking forward to hopefully seeing this manuscript being published in the near future.

Stefan Krause

*** See Nature Research's author and referees' website at www.nature.com/authors for information about policies, services and author benefits.*

REVIEWERS' COMMENTS

Reviewer #1 (Remarks to the Author):

I commend the authors of Nature Communications manuscript NCOMMS-20-22175A for the extremely thorough nature of their revisions to what was already a novel and potentially significant manuscript. I found their responses to the reviewers' comments to be thoughtful and considered, and I appreciate the highly detailed justification of the methodological and analytical steps taken in their work (and additional information in supplementary material). As a result, I feel that this new version of the manuscript satisfies the queries I and other reviewers had in relation to the previous version of the document. I particularly appreciated the streamlined analysis/interpretation section, and the additional detail in the methodology in light of comments on the previous version has helped to clarify questions I previously had. I have noted a few very minor additional comments below (mostly syntax/clarity), but these should not be difficult to address. I reiterate my support of this manuscript, and feel that it will make an excellent contribution to Nature Communications.

Specific minor comments

L27: Consider changing 'lower baseflow score' to 'lower baseflow contribution'

L29-30: Consider changing 'land surface variability' to 'surface climate and land-use variability'

L30-31: This sentence ('Streams without pronounced groundwater signatures...') somehow seems a bit out of step with the rest of the abstract. I suggest removing it, or expanding upon it in more depth to characterise the physiographic settings of streams with shallow and deep groundwater contributions as well.

L37: I suggest replacing 'end member' with 'contributor to', for the sake of clarity.

L86: Suggest replacing 'regarding' with 'of'.

L143: Replace '(refs 8, 38, 39)' with superscript/footnote references.

L159: I'm a bit surprised that sites without groundwater contributions (ie. atmospheric signatures) have a phase lag or greater than a few hours. Can you speculate (briefly) as to what might be causing this?

L215-217: Missing word: 'Thus, landforms and geologic structures ARE likely, in part...'

L228-229: What does 'groundwater connectivity characteristics can be accomplished' mean? I suggest rewording to 'groundwater connectivity characteristics can be ascertained' or similar, if this is what you mean to say.

L230-231: Missing word: 'Because low cost stream temperature MEASUREMENT is currently being performed...'

L245-247: Please convert the imperial units (cfs) to metric, to avoid confusion.

L314: Consider giving the range and std of temperature trend for deep groundwater sites, to allow comparison to atmospheric and shallow groundwater signature locations.

L355: I'm not sure that 'surface temperature' is the best term here? Maybe rephrase and say something like, '...also be particularly vulnerable to the impacts of single season heatwaves?'

Reviewer #2 (Remarks to the Author):

The manuscript explores groundwater contributions to streams, relying on a spatially diverse dataset across the US. The revised manuscript includes thoughtful responses to each reviewer's comments, through which has improved the manuscripts impact to the broader community. I support publication of this revised manuscript. The one suggestion to consider is within the abstract and/or the introduction. Many readers will read the abstract, especially the last sentence, and not understand the importance of this contribution. This may simply be done by providing examples of stressors experiences within freshwater systems (e.g. water availability during droughts to meet demand; new and old water pollution impacting stream health). These are just 2 examples, and while unsure if NComm. allows this, incorporating some statement that explains the "so what" for non-groundwater folks and relating to sustainable water management solutions under changing climate/landuse would increase your overall impact. - Durelle Scott

Reviewer #3 (Remarks to the Author):

The authors have used the revision of their paper to further improve their original manuscript, resulting in an all round well written and highly relevant contribution to the journal. I am satisfied that the comments I made on the previous manuscript have been adequately addressed and I thoroughly enjoyed reading the revised paper.

I particularly appreciate the added discussion of limitations arising from the length of the time series analysed as well as further information provided on the depth criteria used to distinguish water sources.

I made a comment regarding observed Ar numbers which the authors addressed to my satisfaction. They mention additional evidence to their statement being provided in a manuscript of some of the co-authors that is currently under review - should the timing of production of this paper and the other paper under review allow for it, I would recommend cross-referencing to this paper as this would improve the line of evidence, but I wouldn't want to make this a requirement should the timing of both papers make this difficult.

In summary, I compliment the authors to a great contribution that I am absolutely sure will be of great interest to a wide readership of the journal - I certainly enjoyed reading it very much. I recommend publishing it in its current form.

Stefan Krause

REVIEWERS' COMMENTS

We want to reiterate our thanks to the three reviewers for their support of our manuscript, and their comments and suggestions that substantially improved our revised version. We address the reviewers' additional comments below.

Reviewer #1 (Remarks to the Author):

I commend the authors of Nature Communications manuscript NCOMMS-20-22175A for the extremely thorough nature of their revisions to what was already a novel and potentially significant manuscript. I found their responses to the reviewers' comments to be thoughtful and considered, and I appreciate the highly detailed justification of the methodological and analytical steps taken in their work (and additional information in supplementary material). As a result, I feel that this new version of the manuscript satisfies the queries I and other reviewers had in relation to the previous version of the document. I particularly appreciated the streamlined analysis/interpretation section, and the additional detail in the methodology in light of comments on the previous version has helped to clarify questions I previously had. I have noted a few very minor additional comments below (mostly syntax/clarity), but these should not be difficult to address. I reiterate my support of this manuscript, and feel that it will make an excellent contribution to Nature Communications.

We greatly appreciate your time and effort providing supportive and constructive reviews. We feel your input has greatly improved the presentation of our research. Thank you.

Specific minor comments

L27: Consider changing 'lower baseflow score' to 'lower baseflow contribution'
This edit has been made.

L29-30: Consider changing 'land surface variability' to 'surface climate and land-use variability' After review, we agree this phrase is not well suited here, and have restructured the abstract to better reflect our intent.

L30-31: This sentence ('Streams without pronounced groundwater signatures...') somehow seems a bit out of step with the rest of the abstract. I suggest removing it, or expanding upon it in more depth to characterise the physiographic settings of streams with shallow and deep groundwater contributions as well. This sentence was revised.

L37: I suggest replacing 'end member' with 'contributor to', for the sake of clarity. This edit has been made.

L86: Suggest replacing 'regarding' with 'of'. This edit has been made.

L143: Replace '(refs 8, 38, 39)' with superscript/footnote references. This edit has been made.

L159: I'm a bit surprised that sites without groundwater contributions (ie. atmospheric signatures) have a phase lag or greater than a few hours. Can you speculate (briefly) as to what might be causing this? As the standard deviation of this mean atmospheric category phase lag is larger than the value itself, we added the text '*...that is not significantly different than zero phase lag.*' We believe annual signal phase lags on the order of a few days result simply from imprecision in the paired air/water temperature signal processing when not using streamside air temperature as discussed regarding negative phase lags the methodology section (lines 524 - 526). We added text here to reflect this imprecision could also induce multi-day atmospheric signature phase lags. Also, our categorization values are not mutually exclusive, rather the analysis derives which signature is dominant (lines 173-175).

L215-217: Missing word: 'Thus, landforms and geologic structures ARE likely, in part...'. This edit has been made.

L228-229: What does 'groundwater connectivity characteristics can be accomplished' mean? I suggest rewording to 'groundwater connectivity characteristics can be ascertained' or similar, if this is what you mean to say. Good catch, we replaced 'accomplished' with 'inferred'

L230-231: Missing word: 'Because low cost stream temperature MEASUREMENT is currently being performed...'. This edit has been made.

L245-247: Please convert the imperial units (cfs) to metric, to avoid confusion. This edit has been made.

L314: Consider giving the range and std of temperature trend for deep groundwater sites, to allow comparison to atmospheric and shallow groundwater signature locations. The following text has been added to line 341-342: "The six sites with deep GW signatures had rates of warming ranging from 0.01 to 0.05 °C yr⁻¹ (μ : 0.01 °C yr⁻¹)."

L355: I'm not sure that 'surface temperature' is the best term here? Maybe rephrase and say something like, '...also be particularly vulnerable to the impacts of single season heatwaves'? This text has been modified.

Reviewer #2 (Remarks to the Author):

The manuscript explores groundwater contributions to streams, relying on a spatially diverse dataset across the US. The revised manuscript includes thoughtful responses to each reviewer's comments, through which has improved the manuscripts impact to the broader community. I support publication of this revised manuscript. The one suggestion to consider is within the abstract and/or the introduction. Many readers will read the abstract, especially the last sentence, and not understand the importance of

this contribution. This may simply be done by providing examples of stressors experiences within freshwater systems (e.g. water availability during droughts to meet demand; new and old water pollution impacting stream health). These are just 2 examples, and while unsure if NComm. allows this, incorporating some statement that explains the “so what” for non-groundwater folks and relating to sustainable water management solutions under changing climate/landuse would increase your overall impact. - Durelle Scott

Thank you for drawing our attention to this missed opportunity in the abstract. We have reworked the 2nd half of the abstract to better reflect the “so what” of our presented research, particularly the importance of delineating locations of shallow groundwater contributions. We greatly appreciate your time and energy creating meaningful reviews as they have helped strengthen the research within our current manuscript.

Reviewer #3 (Remarks to the Author):

The authors have used the revision of their paper to further improve their original manuscript, resulting in an all round well written and highly relevant contribution to the journal. I am satisfied that the comments I made on the previous manuscript have been adequately addressed and I thoroughly enjoyed reading the revised paper.

I particularly appreciate the added discussion of limitations arising from the length of the time series analysed as well as further information provided on the depth criteria used to distinguish water sources.

I made a comment regarding observed Ar numbers which the authors addressed to my satisfaction. They mention additional evidence to their statement being provided in a manuscript of some of the co-authors that is currently under review - should the timing of production of this paper and the other paper under review allow for it, I would recommend cross-referencing to this paper as this would improve the line of evidence, but I wouldn't want to make this a requirement should the timing of both papers make this difficult.

In summary, I compliment the authors to a great contribution that I am absolutely sure will be of great interest to a wide readership of the journal - I certainly enjoyed reading it very much. I recommend publishing it in its current form.

Stefan Krause

Thank you for your kind words and support of this revised manuscript, which has been aided by your thoughtful and thorough reviews throughout the process. At this time, we are able to cross-reference one of the two papers mentioned in the previous review, as it was just published this week in Ecological Indicators. Text has been added to the methods section line 461-462, and the citation has been added.